# Review of Label-Free Monitoring of Bacteria: From Challenging Practical Applications to Basic Research Perspectives

**DOI:** 10.3390/bios12040188

**Published:** 2022-03-22

**Authors:** Beatrix Péter, Eniko Farkas, Sandor Kurunczi, Zoltán Szittner, Szilvia Bősze, Jeremy J. Ramsden, Inna Szekacs, Robert Horvath

**Affiliations:** 1Nanobiosensorics Laboratory, Centre for Energy Research, Institute of Technical Physics and Materials Science, 1121 Budapest, Hungary; farkas.eniko@ek-cer.hu (E.F.); kurunczi.sandor@ek-cer.hu (S.K.); szittner.zoltan@ek-cer.hu (Z.S.); szekacs.inna@ek-cer.hu (I.S.); 2MTA-ELTE Research Group of Peptide Chemistry, Eötvös Loránd Research Network (ELKH), Institute of Chemistry, Eötvös Loránd University, 1120 Budapest, Hungary; szilvia.bosze@ttk.elte.hu; 3National Public Health Center, 1097 Budapest, Hungary; 4Clore Laboratory, Department of Biomedical Research, University of Buckingham, Buckingham MK18 1AD, UK; jeremy.ramsden@buckingham.ac.uk

**Keywords:** bacteria detection, bacterial adhesion, biosensors, limit of detection, antibiotic resistance, repellent surfaces

## Abstract

Novel biosensors already provide a fast way to detect the adhesion of whole bacteria (or parts of them), biofilm formation, and the effect of antibiotics. Moreover, the detection sensitivities of recent sensor technologies are large enough to investigate molecular-scale biological processes. Usually, these measurements can be performed in real time without using labeling. Despite these excellent capabilities summarized in the present work, the application of novel, label-free sensor technologies in basic biological research is still rare; the literature is dominated by heuristic work, mostly monitoring the presence and amount of a given analyte. The aims of this review are (i) to give an overview of the present status of label-free biosensors in bacteria monitoring, and (ii) to summarize potential novel directions with biological relevancies to initiate future development. Optical, mechanical, and electrical sensing technologies are all discussed with their detailed capabilities in bacteria monitoring. In order to review potential future applications of the outlined techniques in bacteria research, we summarize the most important kinetic processes relevant to the adhesion and survival of bacterial cells. These processes are potential targets of kinetic investigations employing modern label-free technologies in order to reveal new fundamental aspects. Resistance to antibacterials and to other antimicrobial agents, the most important biological mechanisms in bacterial adhesion and strategies to control adhesion, as well as bacteria-mammalian host cell interactions are all discussed with key relevancies to the future development and applications of biosensors.

## 1. Introduction

Novel sensor technologies offer high-throughput and sensitive kinetic measurements on biomolecules and whole cells [1]. Traditionally, chemical and biochemical sensors are simply used to measure the concentration of a given analyte in an investigated sample. It was, however, then found that the novel sensor technologies could be used to investigate the deposition of particles [2,3], and the formation of adsorbed protein layers [4,5,6,7]. Further work led to the novel sensors being used to investigate cell adhesion and spreading [8,9], and today, these novel sensors offer the possibility to investigate important biological processes in real time, in a completely label-free manner [10]. This possibility has already been found to have applications in research with mammalian cells. For example, basic adhesion processes and signaling events could be investigated with precision and sensitivity beyond the reach of traditional methods. Yet, these novel sensing technologies are still relatively unexploited in research and development involving bacterial cells.

Bacteria are microscopic organisms showing different morphologies, including cylindrical (bacilli), spherical (cocci), comma-shaped (vibrio), and spiral-like (spirilla or spirochetes), inter alia. They vary in size—typically 0.2–2 µm in diameter and 3–5 µm in length. Most bacteria have a cell wall providing shape and rigidity enveloping their lipid membrane. The cell wall consists of mainly peptidoglycan (PG), a polymer matrix comprising cross-linked chains of glucose derivatives, *N*-acetylglucosamine, and *N*-acetylmuramic acid. Bacteria that take up the dye crystal violet are called Gram-positive (after Hans Christian Gram developed the staining technique in 1884) and if they do not, Gram-negative. Gram-positive bacteria have a much thicker PG layer. Pathogenic Gram-positive bacteria include *Streptococcus*, *Staphylococcus*, *Bacillus*, and *Clostridium*. Gram-negative have a thinner PG layer between inner and outer membranes, and in addition to proteins and phospholipids have lipopolysaccharides (LPS) anchored into the outer membrane and projecting into the environment. LPS is made up of three different components: lipid A acting as an endotoxin, the O-antigen triggering an immune response in an infected host, and the core polysaccharide. Gram-negative bacteria cause many serious infections: *Neisseria gonorrhoeae*, *Neisseria meningitides*, *Legionella*, *Haemophilis influenzae*, *Escherichia coli*, *Salmonella*, and *Shigella* are a few examples of the pathogenic members of this class.

Dielectric properties of *E. coli* have been studied by a three-shell spheroidal model [11]. Based on the results of Bai et al., the specific capacitance of the inner membrane is 0.6–0.7 μF/cm^2^ [11]. The conductivity of the periplasmic space is 2.2–3.2 S/m, and the relative permittivity and the conductivity of the cytoplasm are approximately 100 and 0.22 S/m, respectively [11]. In the study by Hölzel, the periplasmic space was found to increase with the square root of the medium ionic strength and they showed that heat treatment of the cells leads to a reduction of cytoplasmic conductivity to 0.9 mS/cm, likely due to the efflux of ions through the permeabilized inner membrane, measured by electrorotation [12].

Some bacteria—belonging to the phylum Tenericutes—lack a cell wall; many members of this phylum are pathogens. Mycobacteria belong to the diverse family of Actinobacteria and the main components of their cell wall are the PG layer, mycolic acid, and arabinogalactan. Mycobacteria are Gram-positive organisms, but their envelopes share notable features with Gram-negative cell walls, having a thin PG layer and an outer permeability barrier acting as a pseudo-outer membrane [13,14]. Bacterial cell wall components are promising targets for antibiotic/antimicrobial compound discovery. Gram-positive and Gram-negative bacteria respond to different antibiotics and antimicrobial drugs.

Classically, pathogenic bacteria are classified as extracellular, facultative intracellular, and obligate intracellular. There is growing evidence that extracellular bacteria can survive inside host cells (mainly macrophages) where they hide from the immune system, survive in active or latent forms over prolonged intervals of time, and even proliferate. These bacteria lead to some of the most insidious infections that threaten human health and welfare globally. The main factors that contribute to this are: (i) rising antimicrobial resistance; (ii) increasing number of immunosuppressed patients; and (iii) healthcare-associated infections. The risk of developing an active disease is obviously higher among persons with compromised immune systems (elderly and transplant patients, HIV-positive cases, people with diabetes, cancer, autoimmune diseases, etc.). Intracellular infections are especially difficult to eliminate because the bacteria employ some ingenious mechanisms. These bacteria mainly reside inside phagocytic cells, but also find their way into non-phagocytic cells (epithelial cells, hepatocytes, fibroblasts, etc.). The host cells are not only primarily infected but also act as a reservoir for the bacteria, which can disseminate themselves in other tissues, leading to systemic infection. It has been proposed that bacterial pathogens should be labeled as exclusive extracellular, dual intracellular/extracellular or exclusive intracellular based on their infective lifestyle in the host [15].

Bacterial adhesion is the initial step in colonization and biofilm formation. Biofilms can be useful in environmental technologies and bioprocesses (for example, production of antibiotics, fermentation of food products, help in digestion); however, they can also cause infection, slime formation and pathogen contamination [16].

On the surface of pathogenic bacterial cells, one can usually find surface-associated molecules, called adhesins, that recognize eukaryotic surface components and mediate tight association with host tissues [17]. Adhesive molecules that also promote internalization of the microorganism are called invasins [17]. Bacterial adhesin and invasin recognize the host receptor, which determines the organ and tissue specificity and thus the entry site and mode of colonization as well [17]. Numerous adhesins and invasins can bind to multiple extracellular matrix (ECM) components and to transmembrane receptor molecules such as integrins [17]. Generally, bacteria do not distinguish between immobilized, soluble, and tissue-specific derivatives of ECMs [17]. 

The intracellular lifestyle provides certain advantages to pathogen bacteria: they are protected from the host immune system and antibiotics, they do not have to actuate energy-consuming adherence mechanisms to remain at the site of infection, they can access many nutrients in the cell cytoplasm, and they are likely to find good, stable physical conditions for growth [17,18,19]. Active invasion means that the bacterium can enter and diffuse into deeper subepithelial tissues to survive and systemically disseminate themselves in their hosts [17].

The glycocalyx is a mass of tangled fibers of LPS, other polysaccharides including branched sugar molecules that protrude from the surface of bacteria, surrounding a single cell or a colony of cells [18]. Glycocalyx-mediated adhesion determines the locations of bacteria in natural environments and the formation of biofilms (for example, tooth plaque). Glycocalyx fibers conserve and concentrate digestive enzymes released by the bacteria and directed against the host cell [18,19].

In modern times, a lot of illnesses caused by pathogen bacteria can be medicated due to antibiotics, and antimicrobial and bacteria-repellent surfaces can be also created to reduce contamination [20]. However, during continuous exposure to antibacterial drugs, sequential chromosomal mutations can occur, leading to the appearance of resistance mechanisms step by step [21]. Nevertheless, some photocatalytic antimicrobial surfaces generate oxidizing radicals that do not engender resistance mechanisms [22].

Pathogenic bacteria and antibacterial substances are important targets for identification and detection in various fields, including food safety, security, medicine, forensic medicine, and public health [23]. Conventional laboratory-based methods generally have long processing times (take days or weeks to complete), and require specialized equipment and professional users, hence these methods are costly. Identification of a pathogenic species is typically performed by isolating and culturing on chromogenic agar, followed by Gram staining, microscopic examination, and other biochemical and molecular-biological techniques [24,25]. Each technique has its strengths and limitations. Staining is simple and provides valuable information about bacterial morphology. Biochemical assays include detection of enzymes specific to a certain bacterium. Immunological tests include agglutination assays [26] and enzyme-linked immunosorbent assays (ELISAs) [27]. The polymerase chain reaction (PCR) allows multiple copies of a bacterial genome to be made (this process is sometimes called ‘DNA amplification’), hence even just one bacterium could be identified by genome sequencing after PCR [23,28]. However, contamination and wrong DNA pairing can cause false-positive or -negative results, and genetically mutated strains may escape the correct probe matching [29]. The overall technique is still quite time-consuming and costly. Real-time PCR (RT-PCR) analysis can be performed faster, within several hours, but needs special reagents and equipment [23,30]. Mass spectrometry-based methods are also currently widely used due to their speed, high-throughput, and sensitive and specific analysis [31]. The development and implementation of real-time and more sensitive diagnostic methods remains extremely important.

Biosensors provide a direct way of characterizing the adhesion of whole bacteria or parts of them. These devices integrate a biological recognition element (e.g., whole cells, receptors, enzymes, antibodies) with a signal transduction element. In the most advanced systems, additional labels are not needed for signal generation and the information is provided in real time. Optical and amperometric techniques are the most commonly used for signal transduction [23]. Antibiotic resistance has also been studied with label-free biosensors [32,33].

In this review, in the first part, we summarize the most successful mechanical, optical, and electrical biosensors in detecting bacteria, and we also discuss monitoring of antibiotic resistance by novel, label-free techniques. After, we provide a brief summary of biological processes potentially interesting for label-free monitoring. From this perspective, we discuss bacterial adhesion, adhesive properties and mechanisms of bacterial adhesion, the role of their glycocalyx in the process, and even their entry into mammalian host cells.

## 2. Applications of Biosensors in Bacteria Detection

Novel biosensor technologies offer sensitive kinetic measurements in real-time. Some of them are label-free, so labels and dyes are not required for performing the measurements. Therefore, these assays can be cheaper and less time consuming than conventional label-based endpoint measurements. Their present applications in microbiological experiments include exploring the presence of the bacteria and recognizing them, and measuring their concentration in the sample of interest (for example in food products, body fluids). In healthcare, these devices will be more and more important due to capability of sensing resistant bacteria as well, which represents a serious and challenging problem today. Creating bacteria repellent surfaces is a novel and very useful technology in minimizing contamination, for instance in hospitals and public transport vehicles. The repellent effect of these surfaces can be easily tested with label-free biosensors. In this main section, we summarize the types of novel biosensors, their limit of detection, and assay time in measuring bacteria, and we discuss the research challenges as well.

### 2.1. Recognition of Bacteria Using Biosensors

Timely detection and identification of pathogenic bacteria is very important for different fields including medicine, food safety, and security [23,34,35]. Infectious illnesses cause millions of deaths and hospitalizations each year. *Salmonella*, *Campylobacter*, *Enterohaemorrhagic Escherichia coli*, *Listeria*, and *Vibrio cholerae* are the most common cause of bacterial food- and water-borne infection according to the World Health Organization [23], and *Staphylococcus aureus* is one of the most common etiological agents in hospital-acquired infections [35]. As discussed above, conventional laboratory-based methods for bacterial detection and identification typically require specialized equipment, often non-automated, have long processing times, are costly, and some of them lack sensitivity and specificity. On-site identification of bacteria and expected response to antibiotic treatment may be critical for clinical diagnosis and treatment. Application of biosensors for bacterial detection has been summarized in a recent book [36]. In the coming sections, we shall highlight some important points from earlier research and review the most important new developments.

Biosensors provide a cost-effective, rapid, and sensitive method of detection. Recent strategies for detection of bacteria involve biosensor techniques based on recognition of bacterial components or whole bacterial cells. Nucleic acid-, intracellular protein- (enzymes), and bacteria-produced exotoxin-based biosensors include sample preparation steps to achieve target bacterial component separation by lysis or other disruption [23,37,38,39,40]. The main disadvantage of these methods is the necessity of component isolation and purification, which raises the cost and time of these measurements. Thus, biosensors for the direct detection of whole bacteria is preferable for point-of-care diagnostic. The main challenge in biosensor detection is to obtain a concentrated sample from a high-volume flowing environment. This is critical because the infectious dose of bacteria for many human pathogens is remarkably low [23]. The concentration of bacteria can be achieved through centrifugation or filtration of the sample, concentration by using nanoparticles, and some biosensor techniques have the possibility to concentrate the target cells, for example with electrodes [41]. Dielectrophoresis is often used in biosensors employing microfluidics [42,43,44].

A bioreceptor provides the specificity of the analytical determination in the biosensor technique. The surface of the transducer must be modified with any molecule that bacteria of interest are known to bind to. It can be mono- or polyclonal recombinant antibodies [45], glycans and lectin proteins [46], and bacteriophage [47]. Bacteria have different surface antigens presented on the cell or the flagellae, such as surface proteins, glycoproteins, peptidoglycans, and lipopolysaccharides that can act as targets for biorecognition. Antibodies are the most widely used bioreceptors. Aside from their high specificity, their production and purification is time-consuming and costly, cross-reactivity may occur, their proper orientation on the sensor surface is difficult to achieve, and they are generally rather unstable [23]. Bacteria exhibit many epitopes that can also cause so-called nonspecific interactions with the unmodified sensor surface. However, appropriate design of the surface can be exploited to achieve effective differential binding of bacteria [23]. Improvement of the detection limits (signal-to-noise ratio) for bacteria is a key assignment for the next generation of biosensor techniques. Figure 1 schematically shows possible ways to recognize bacteria and sense antibiotic resistance using biosensors.

#### 2.1.1. Optical Biosensors

Optical biosensors utilize the changes in the optical properties of the sensor surface caused by the bound analyte, and these alterations are then transduced to a detector [23,48,49]. The mentioned methods are often divided into two categories, fluorescence- or nanoparticle-based labeling techniques or label-free ones [23]. Fluorescence-based biosensors can have high sensitivity [50,51,52,53,54,55,56]; for example, Mouffouk and co-authors applied a fluorescent dye-loaded micelle system to detect 15 cells/mL of *E. coli* [23,57]. Colorimetric biosensors recently appear to be attractive for the researchers in the field of bacteria sensing due to their simplicity and rapidity (color reactions for detection in few minutes) [58,59,60,61,62]. These types of sensors can be used in point-of-care diagnostic platforms in hospitals and to improve the control of health-risks associated with contaminated food consumption [35].

However, the major disadvantage of using label-based optical biosensors is the requirement for sample labeling with the fluorescent reagents, which adds time and cost to the assay [23], furthermore, the dyes and labels may disturb the normal physiology of the cells [63].

Among label-free optical biosensors, surface plasmon resonance (SPR) has been employed for the detection of a range of materials since the first commercially available appliance was released in 1990 by Biacore [23]. SPR systems contain a source of plane-polarized light which then passes through a glass prism, its surface contacts the bioreceptor-functionalized transducer interface, which is generally a thin gold film [23,64,65]. The binding of the material of interest to the transducer surface changes the local refractive index, which in turn alters the angle of light leaving the prism (the so-called “SPR angle”) [23].

Approximately 50% of the biosensors used in the world are based on the SPR technique, which similarly dominates the antibiotic biosensing field [21]. These biosensors have been developed for the detection of whole bacterial cells using a variety of bioreceptors, for instance lectins [66], molecularly imprinted polymers [67], antimicrobial peptides [68], mucin [69], antibodies [70,71,72,73,74,75], and bacteriophage [76,77]. Unfortunately, the detection of whole bacteria using this technique has a low sensitivity compared to other methods, due especially to the small difference in refractive index between the bacterial cytoplasm and the aqueous medium, and the limited penetration of the electromagnetic field into the bacteria, which is what controls the signal transduction in the biosensor [23,78]. In recent years, hybrid configuration SPR biosensors have been made by combining metals and dielectrics [79,80]. In localized surface plasmon resonance (LSPR), metal nanoparticles are used to intensify the sensitivity of the system [23]. Fiber optic probes enhance sensitivity, too [68,81]. Some other strategies to enhance the sensitivity of SPR-based bacterial sensors include transducer surface modifications, e.g., using nanorods [82], and sandwich-type assays with nanoparticles for analyte capture to boost the signal [23]. The use of gold-coated magnetic nanoparticles increased the sensitivity down to 3 CFU/mL for *E. coli* [78]. However, LSPR is reported to be less sensitive and sometimes limited by a sample with biological matrix [23,83,84]. SPR can be improved by imaging methods (iSPR) to get more details on the antibiotic binding process [21].

Surface-enhanced Raman scattering (SERS) improves the amplitude of a Raman spectrum manifold and has been applied in combination with other methods to detect bacterial cells in blood medium [23,85]. SERS (surface-enhanced Raman spectroscopy) is a very sensitive technique [82,85,86]; it is able to distinguish different bacteria down to the single bacterium level [87,88]. Especially with these enhancing strategies, SPR-based systems are still large, costly appliances which have not yet been adopted for point-of-care diagnostics [23]. Although Spreeta SPR chips (Texas Instruments Inc., Dallas, TX, USA) have permitted the development of a miniaturized SPR-based biosensor, this still needs a microfluidic system and is thus confined to the research laboratory [23].

Adányi et al. developed an OWLS-based bacterial biosensor containing silicate in modified bacteria for the detection of stressors, environmental pollutants, and antibiotics [89]. Horvath et al. utilized the optical waveguide lightmode spectroscopy (OWLS) sensor for online monitoring of the adhesion of *E. coli* K12 cells to the sensor surface with a novel waveguide design [90]. The so-called reverse symmetry waveguide employs waveguide substrates with refractive index lower than that of the aqueous cover media. This results in an increased and fine-tunable penetration depth of the monitoring evanescent optical field, fitted this way to the size of living cells [10,91,92].

Metal-clad waveguides also provide an enhanced evanescent field compared to conventional waveguide sensor designs [93,94]. Such configurations are also termed metal-clad leaky waveguides (MCLW), where the increased sensitivity is due to the thin metal layer (8.5 nm) between substrate (glass) and spacer (silica) [95]. The design of the sensor allows the detection of whole bacteria (1–5 μm). The limit of detection is about 1 × 10^3^ bacterial spores/mL [95,96,97,98,99].

Reflectometric interference Fourier transform spectroscopy (RIFTS) with porous silicon is an alternative way to detect bacteria. Yaghoubi et al. detected *E. coli* and *S. aureus* in a real-time mode with a limit of detection (LOD) of about 10^3^ cells/mL [100]. Another novel biosensing technique is the nanophotonic interferometric biosensor which provides a fast method for the identification of nosocomial pathogens for the diagnosis of infections [101].

One general problem with most of the biosensor assays (as well as the mechanical and electrochemical ones discussed in the following sections) is that quantification of the number of bacteria detected by the sensor relies on calibration, which in turn relies on sometimes questionable assumptions, although this might still be better than having to use labels. The elegant approach to quantifying particles, including protein molecules, that is uniquely available for OWLS [2,3,7,102] cannot be applied to bacteria because they are too large, nor can the different, also elegant, approach to quantifying eukaryotic cell number and shape [8,9,10] be applied to bacteria because they are too small. Yeh and Ramsden devised a heuristic solution to the problem [49] that can be applied when kinetic information is available.

#### 2.1.2. Mechanical Biosensors

The two main types of mechanical biosensors are based on quartz crystal microbalance (QCM) or cantilever technology [23]. QCM sensors are label-free piezoelectric biosensors which detect the resonance frequency change caused by the increased mass of the binding material on the sensor surface [23,65,103]. QCM sensors are able to detect bacteria as well and apply a variety of bioreceptors (for instance, aptamers [104,105,106], DNA [107], antibodies [108,109,110], or molecularly imprinted polymers [111,112]). The most sensitive sensors among them are the aptasensors [104,105] which can detect even bacterial spores [111]. Due to the development of sandwich-type assays, which apply nanoparticles for signal amplification, the detection of very few bacterial cells have been accomplished, even down to 10 CFU/mL in some cases as Salam and co-authors reported [23,113]. For the detection of *E. coli*, Yu and colleagues prepared an aptamer functionalized QCM surface, with pre-enrichment with antibodies immobilized magnetic nanobeads [104]. This aptasensor was able to detect 1.46 × 10^3^ CFU/mL of *E. coli* within 1 h. Gold nanoparticles can be used to increase the sensitivity of QCM analysis by conjugating to detection probe antibodies [104]. Masdor et al. reached a DL of 150 CFU/mL for *Campylobacter jejuni* [108].

Microcantilever sensor technology offers high sensitivity, rapid response times, and it can be miniaturized for the development of point-of-care sensors. It typically has a bioreceptor functionalized microcantilever which oscillates at a certain resonant frequency. Because of the induced mechanical bending, the resonant frequency of the cantilever changes upon an increase in mass on the sensor surface [23]. Microcantilever and nanocantilever sensors can detect even individual cells in a short time (30 min) [114,115,116]. The piezoelectric-excited millimeter-size cantilevers (PEMC) using antibodies as bioreceptors have been able to detect one hundred *Listeria monocytogenes* cells in milk [117], and one *E. coli* cell in buffer [23,118].

#### 2.1.3. Electrochemical Biosensors

Electrochemical methods (~21%) are the second most applied detection principle after the SPR (surface plasmon resonance) technique [21]. They are dominated by voltammetric and amperometric methods, but these types of sensors include impedimetric and potentiometric sensing techniques as well [23]. These sensors analyze the impedance of bacterial cells when they are attached to or associated with the electrodes. Electrochemical methodologies have lower manufacturing costs and ease of system miniaturization and integration. Electrochemical biosensors, more specifically, impedimetric sensors, can take the leading position in this area, however, the optimization, miniaturization, and clinical trials need to be performed before any item is sent into the market [23].

Among others, their advantages are the potential use in point-of-care testing thanks to the low cost and miniaturizability [23].

For detecting whole bacteria, impedimetric biosensors are a very good choice because they are relatively cheap and highly sensitive label-free systems. Furthermore, they are easy to miniaturize, so they may be used as point-of-care systems. In these types of biosensors, the analyte–bioreceptor interaction causes a change in electron transfer resistance and capacitance across a working electrode surface. The analyte binding raises with higher concentration, thus, the impedance across the electrode surface changes and it is detected by a transducer. The impedance increases or decreases depending on the analyte [23]. In general, bioreceptors are antibodies [119,120,121] and aptamers [122,123], but they may be other molecules as well which are able detect proteins or whole bacteria and viruses [23,124,125]. The main advantage is that there are no requirements for the analyte to be an enzymatic substrate or for formation of electroactive species. However, no impedance biosensors have had widespread commercial success, doubtless due to disadvantages like irreproducibility, high limits of detection, and problems with nonspecific binding [23].

In case of whole bacterial detection, viable *E. coli* cells in mixed populations of living and dead cells have been reported by de la Rica and co-workers [23,126]. Differentiating living cells from dead cells can be beneficial when the number of viable cells shows the true pathogenic count. In general, living cells are voluminous compared to dead cells. The interference with the electric field is higher in case of the viable cells than that of the dead cells due to the higher cell volume of living bacteria, and this difference can be detected by impedance and capacitance measurements [23,126]. Robotic layer-by-layer construction and miniaturization of these techniques may develop sensor performance with sufficiently high reproducibility for commercialization [23].

Voltametric biosensors measure the current change beside the applied potential caused by the analyte. The advantages of this technique are rapid detection and high sensitivity, but the analysis and setups are complicated [127]. Bioreceptors used in this technique are antibodies [128], bacteriophage [129], aptamers [130], enzymes [131], and peptides [132] for bacterial cell detection.

Potentiometric biosensors apply ion-selective electrodes to examine the potential of a solution based on specific interactions with ions in the solution. This technique measures the change in potential that occurs upon analyte interaction at the working electrode [23]. Most of the potentiometric biosensors include membrane-based ion-selective electrodes (ISEs) and ion-selective field-effect transistors (ISFETs) [133]. Potentiometry is widely used in the biosensor field; however, potentiometric biosensors for the detection of whole bacterial cells are few. This is probably because potentiometry cannot give specific signals for large analytes such as bacteria (a false positive signal may arise as a result of a nonspecific ion flux) [23]. This can be alleviated by the immobilization of biorecognition ligands on the sensor membrane. For example, a biocompatible interface was developed by Silva and co-workers for the detection of *Salmonella typhimurium* [134]. Silva et al. developed a sensitive and low-cost disposable paper-based potentiometric immunosensor to detect food-borne pathogens (e.g., *S. typhimurium*) [134].

Amperometric biosensors are based on direct measurement of the current generated by the reduction or oxidation of species created by the analyte–bioreceptor interaction [23]. The bioreceptor is usually an enzyme, for instance glucose oxidase [23], but antibodies are also used [135]. The generated current is directly proportional to the analyte concentration [23]. The advantages of these biosensors are their relative simplicity, excellent sensitivity, and miniaturizability [23]. The disadvantage is the low specificity, depending on the used potential [23]. Although in the field of biosensing, amperometry is the most common detection method, in case of whole-cell bacterial detection, this technique is not widely applied [23]. Eissa and Zourob, an active group in this field, reported an electrochemical biosensor which was composed of an array of gold nanoparticle-modified screen-printed carbon electrodes on which magnetic nanoparticles coupled to specific peptides were immobilized via streptavidin-biotin interaction. Taking advantage of the proteolytic activities of the protease enzymes produced from two bacteria on the specific peptides, the detection was achieved in 1 min [132]. The achieved limit of detection was 9 CFU/mL for *L. monocytogenes* and 3 CFU/mL for *Staphylococcus aureus*. That paper presents a useful tabulation of electroanalytical methods for the determination of Listeria monocytogenes and *S. aureus*, showing their main characteristics, among others’ detection limits, which fall in the range of 1–100 CFU/mL. As it is well known, the sensitivity of amperometry can be enhanced by the application of mediator molecules, which amplify the response current generated at the electrode. An amperometric biosensor has been developed for the *Streptococcus agalactiae*, where signal amplification was achieved by streptavidin conjugated horseradish peroxidase (HRP) [136]. This made possible the detection of *S. agalactiae* in the range of 10^1^ to 10^7^ CFU/mL. Due to stability issues, the general use of HRP in amperomteric sensors is limited.

The whole bacteria detecting biosensors, their obtainable LOD values and assay time are summarized in Table 1.

To conclude and summarize this section, label-free optical biosensors have certain advantages, i.e., relatively simple measurement procedures, application of parallel sensor units, and thus high-throughput monitoring capabilities [165,166,167,168,169]. Usually, optical biosensors are less sensitive to vibration and electrical noise. The disadvantage of the evanescent field-based methods is that if the sample does not reach the evanescent field (approximately a 100–150 nm thick layer above the sensing surface), there is no biosensor signal at all [165,166,170], therefore, sample concentration is sometimes critical. With using labels in case of label-based optical biosensors, the sensitivity can be significantly increased, however, the labels may disturb the cells and the measurements can be more complicated and time-consuming, and the labels increase the costs of the assay [63,166].

In case of mechanical biosensors, the advantage is that mechanical parameters can be easily extractable and high sensitivity can be achieved, opening up novel directions in the rapidly growing field of mechanobiology. However, as their working principle is usually based on monitoring a resonance frequency, these devices are highly sensitive to any vibration noises present in their environment [65].

The advantages of electrochemical methods are the lower manufacturing costs and ease of system miniaturization and integration, and the ease of use in integrated point-of-care devices [23]. In general, the disadvantages of some systems, which should be critically considered, are the irreproducibility, high limits of detection in real-life assays, and problems with nonspecific response [23,171].

### 2.2. Resistance to Antibacterials and to Other Antimicrobial Agents

Inevitably, antibacterials rapidly lose their effectiveness as bacteria adapt and develop resistance to them; some bacteria even resist multiple drugs [172,173]. Sequential chromosomal mutations can occur during continuous exposure to antibiotics, which lead to the appearance of resistance mechanisms step by step [21]. Various factors can participate in the development of antibiotic resistance: (i) extensive and sometimes unnecessary applications of antibiotics in pet care and agriculture; (ii) overprescribing of antibiotics for humans; (iii) patients not taking antibiotics as prescribed; (iv) the increased presence of antibiotics or their metabolites in the environment; and (v) lack of rapid laboratory tests for the identification of bacteria leading to the application of broad-spectrum antibiotics [21,174]. Although the efficacy of antibiotic treatment can never be permanently guaranteed, the practice of medicine will be greatly helped if the further progress of bacterial resistance can be slowed down. To this end, it will be useful to identify natural compounds or develop new antibiotics that can attack novel target structures or overcome the existing resistance mechanisms. Unfortunately, the development of novel antibacterial compounds is difficult. Until the early 1980s, various antibacterial classes had been identified. After that, until the end of the 20th century, there was no further discovery of truly novel antibacterials [21,175]. Then, in recent years, three novel antibacterials (daptomycin (lipopeptide) [21,176], tigecycline (glycylcycline), and retapamulin (pleuromutilin)) have been validated [21,177,178]. Other accepted antibacterials are merely derivatives of known ones. There are currently only five real novel antibacterial compounds in clinical development [21].

These compounds are extensively used agents against microbial infection in animals as well [179]. However, inappropriate doses may result in antibacterial residues in food of animal origin and possible side effects may influence human health [179]. The accumulation of these chemicals in natural waters, beverages in both their metabolized and unmetabolized forms are of interest, too [180].

Antibiotics and other type of antimicrobial compounds are in some cases probably not able to overcome the binding capacity of the glycocalyx and therefore simply cannot reach their bacterial target [18].

Adhesion has an important role in the success of pathogens, so an effective way to prevent or combat bacterial infection can be the prevention of their adhesion. Three paths to antibiotics that interfere with glycocalyx formation or function in specific pathogens may be:-occupation and blockage of the active site of a lectin that mediates the adhesion of bacterial glycocalyx fibers to the fibers of host cells,-blockage of the “receptor” sites on host cells, the glycoprotein fibers to which bacterial fibers adhere directly, or-disruption of the production of glycocalyx fibers (mimics its normal substrate and therefore occupies the enzyme’s active site) [18].

Vaccines against the surface components of these pathogenic bacteria have been shown to be protective, and a range of vaccines against the pili/glycocalyx are being developed. The ion exchange barrier, the distal polysaccharide portion, influences antibiotic sensitivity. Furthermore, divalent cation concentrations affect bacterial sensitivity to antibiotics. The effect of the glycocalyx is complicated to isolate. In the absence of human and bacterial polysaccharides, examination of the antibacterial effect of a compound using laboratory cultures is hardly predictive [19].

Currently, the main analytical tools for antibiotic detection are liquid and gas chromatographic techniques, mostly combined with mass spectrometric analysis. For example, the detection of antibacterial agents in environmental waters is generally performed by HPLC-MS or HPLC-MS/MS [21]. Other methods, such as electrochemical detection, UV, or fluorescence spectroscopy have minor significance because of their lower sensitivities. In the food industry, milk is one of the most controlled products, because typical antibiotics can easily accumulate in higher amounts due to its lipophilic properties. Because of the widespread consumption of milk, it is essential to control its threshold levels of antibiotics. It is necessary to detect the antibiotics prior to contamination of the food chain; for example, already at the farm [21].

To examine antibiotics, traditional microbiological tests based on the inhibition of bacterial growth can be applied [181]. These methods are simple and cheap; however, they must be carried out in the laboratory circumstances, and they are time-consuming [181,182]. Alternative methods, such as liquid chromatography (LC) coupled to mass spectrometry (LC–MS) or UV (LC–UV) detection and capillary electrophoresis (CE)-based methods are expensive, need careful sample preparation, and can only be performed by trained staff [181,182]. Compared to these analytical methods, biosensors can be used as cost-efficient, rapid, and simple methods that can be applied without further complicated sample preparation. They have been found to be comparable to conventional techniques with respect to sensitivity and specificity of antibiotic detection and thus fulfill international regulatory requirements [21].

Kling and co-workers demonstrated a microfluidic platform enabling the electrochemical readout of up to eight enzyme-linked assays (ELAs) at the same time [181]. They applied highly sensitive biomolecular sensor systems for the detection of tetracycline and streptogramin, the two commonly employed antibiotic classes. The limits of detection (LOD) are determined, with 6.33 for tetracycline and 9.22 ng/mL for pristinamycin. The employed channel material, dry film photoresist (DFR), allows an easy storage of preimmobilized assays with a shelf life of minimum 3 months. Experiments in a medium are presented by the simultaneous detection of the antibiotics in spiked human plasma within a sample-to-result time of less than 15 min [181].

It is well known that fungi can produce antibacterial agents, but the bacteria themselves can be a robust source of novel antibiotics, as well; however, methods for screening large bacterial libraries for novel antibiotic production are required [183]. Recently, Murray and co-workers showed a rapid and quantitative high-throughput liquid culture screening method for antibiotic production by bacteria. This technique screens both mono- and coculture mixtures of bacteria in vitro. The 5-day procedure starts after the colonies are quadrant-streaked from agar plates and allowed to grow for 36–48 h. Then, filter plates with the bacteria of interest are centrifuged and the filtrate treated with pathogenic bacteria. After 24 h incubation, pathogenic inhibition is assessed via optical density (at 590 nm) [183]. Hits are determined as any inhibition at least two standard deviations from the control in tryptic soy broth (TSB) [183]. Strong producers are categorized as those hits that have ≥30% inhibition. Over 260 bacterial species were screened in monoculture, and 38 and 34% were found to produce antibiotics inhibiting of *S. aureus* or *E. coli*, respectively (with 8 and 4% strong producers, respectively). More than 270 cocultures were investigated, and 14 and 30% were found to generate antibiotics inhibiting *S. aureus* or *E. coli*, respectively [183].

The number of biosensor applications in this field has risen since 2006, especially for the detection of antibiotics and bacteria in the environment [21]. In the design of sensors, magnetic nanoparticles can also be incorporated in order to increase their sensitivity in detecting antibacterials [180]. Despite many different antibiotic recognition principles, just a few biosensor techniques were used in the antibiotic field [21], but the recent developments and studies are very promising.

For example, whole-cell biosensors are easy-to-use screening tools for the rapid and sensitive detection of antibiotics [184]. In a recent study by Lautenschläger et al., the authors developed a novel whole-cell biosensor in *Bacillus subtilis*, based on the β-lactam-induced regulatory system BlaR1/BlaI from *S. aureus* [184]. This system can be applied for detecting β-lactams on solid media or in liquid for identifying potential β-lactam producers [184]. Yin et al. used the σM-mediated regulatory system of *B. subtilis* to create a whole-cell biosensor for the detection of cell envelope-acting antibiotics [185]. As a result, the detection range for polymyxin B was between 0.125 and 12 μg/mL [185].

Ates and co-workers tested the microfluidic biosensor (miLab) on the blood, plasma, urine, saliva, and breath samples of pigs who had received piperacillin/tazobactam antibiotics [186]. The result achieved with this device in the plasma of the animals were as accurate as in the standard medical laboratory process [186]. This antibody-free biosensor using penicillin-binding proteins enables a rapid (less than 90 min) and highly sensitive (range of ng/mL) detection of β-lactams [186].

Rawson et al. reported the first-in-human study, where a minimally invasive, microneedle β-lactamase biosensors were used for real-time, in-vivo antibiotic drug monitoring [187]. The limit of detection for the microneedles was 0.17 mg/L for phenoxymethylpenicillin [187]. This experiment can be the first step towards the individualized and automated antibacterial dosing closed-loop control system for human patients [187].

### 2.3. Detection of Resistance Using Biosensors

A fast biosensor for the detection of bacterial growth was developed by Gfeller and co-workers using micromechanical oscillators coated by common nutritive layers [32]. The change in resonance frequency due to the increasing mass on a cantilever array is the basis of the detection scheme [32]. Single bacterial cells and virus particles can be detected in a dry environment with these types of sensors [32,188,189]. The calculated mass sensitivity according to the mechanical properties of the cantilever sensor is 50 pg/Hz, which corresponds to a sensitivity of circa 100 *E. coli* cells [32]. The sensor is able to detect the active growth of *E. coli* cells within 1 h [32]. In addition, this method allows the detection of selective growth of *E. coli* within only 2 h by adding antibiotics such as kanamycin or tetracycline to the nutritive layers [32]. The authors confirmed the growth of *E. coli* by scanning electron microscopy [32]. For this measurement, two cantilevers were coated differently; the growth-inhibiting cantilever was coated by a nutritive layer containing 10 g/mL kanamycin, which inhibits the growth of *E. coli* XL1-Blue [32]. However, the growth-supporting cantilever was coated by the same nutritive layer, but tetracycline (10 g/mL) was added to it [32]. *E. coli* XL1-Blue is resistant to tetracycline and is, therefore, able to grow on such a coating [32]. Both cantilevers were then exposed to equal amounts of *E. coli* XL1-Blue bacteria (1000 cells/cantilever surface) [32]. There was no change in the resonance frequency of the growth-inhibiting cantilevers [32]. On the other hand, the frequency shifts of the growth-supporting cantilevers increased over the first 4 h of the experiment and then reached a constant value [32]. Due to the sensitivity of the sensor, it was possible to detect selective growth of *E. coli* due to antibiotics in less than 2 h [32].

Magnesium zinc oxide (MZO) nanostructure-modified quartz crystal microbalance (MZOnano-QCM) biosensor monitored antimicrobial resistance in case of *E. coli* and *S. cerevisiae* from a small amount of sample (2 mL) within 10 min [190].

Leonard et al. designed biofunctionalized silicon micropillar arrays to provide both a preferable solid–liquid interface for bacteria networking and a simultaneous transducing element that monitors the response of pathogens when exposed to certain antibiotics in real time [191]. The micropillar architecture relays optical phase-shift reflectometric interference spectroscopic measurements as a platform for label-free, culture-free phenotypic antimicrobial susceptibility testing (AST) [191]. Within the interstitial space of the micropillar gratings, cells can freely move and colonize, while colonization on top of the transducing element (as opposed to within) is prevented, thus minimizing losses in optical response [191]. The responses of *E. coli* to different concentrations of five clinically relevant antibiotics (gentamicin, ciprofloxacin, ampicillin, ceftriaxone, sulfamethoxazol-trimethoprim) were optically tracked by PRISM and minimum inhibitory concentration (MIC) values determined and compared to both clinic-based automated AST system readouts and standard broth microdilution testing [191]. The capture of bacteria within these microtopologies followed by incubation of cells with the antibiotic solution yields fast (2–3 h total assay time versus 8 h with automated AST systems) and accurate determinations of antibiotic susceptibility [191] and can also differentiate between bactericidal and bacteriostatic antibiotics [191].

In the study by Obaje and co-workers, the authors presented the manufacturing and characterization of a low-cost carbon screen-printed electrochemical sensor on a ceramic substrate [33]. Applying label-free electrochemical impedance spectroscopy (EIS), the sensor is applicable for detection of blaNDM, which is one of the main antimicrobial resistance factors in carbapenem-resistant Enterobacteriaceae. The sensor was used for the sensitive and specific detection of synthetic blaNDM targets with a detection limit of 200 nM [33]. Its properties proved the suitability of the new electrode materials and manufacturing for further point-of-care test development [33]. The blaNDM gene encoding the NDM beta-lactamase (New Delhi metallo-beta-lactamase) has been identified as having an important role because organisms containing blaNDM are probably multiresistant and only susceptible to “last option” treatments like colistin and tigecycline, which have uncertain efficacies [33,192].

Other selective electrochemical techniques exist for detecting methicillin-resistance from 57 fM genome DNA [193]. Saucedo and co-workers developed a polymer layer on an electrochemical sensor, which can detect antibiotic-resistant bacteria after 4 mg/mL antibiotic treatment (kanamycin, tetracycline, and erythromycin) [194]. A low-cost, sensitive, and rapid (5 min) thin-film transistor nanoribbon sensor can detect cephalosporin- and carbapenem-resistant bacteria [195]. Kumar and co-workers developed a rapid (5 min), sensitive, and miniaturized graphene field effect transistor, with improved detection limit and time [164].

Among optical biosensors, the ultrasensitive nanophotonic bimodal waveguide interferometer is able to detect β-lactamase and carbapenemase coding gene with a detection limit of 28 aM. [161]. A SERS-based method was also claimed to have 100% accuracy identifying methicillin-resistant bacteria [161]. LSPR is shown to sense cephalosporin, ceftazidime, ceftriaxone, and cefotaxime resistance more effectively than chromatography coupled mass spectroscopic techniques [196]. Rapid (30 min) and specific bead-based biosensing can detect ceftadizime and levofloxacin resistance in different bacterial strains [197].

Electro-photonic traps can be also used for bacteria monitoring even at a single-cell level [198,199,200] and resistance can be studied, too [201]. While the bacteria are trapped by optical tweezers based on photonic crystal cavities, antibiotics can be added to the medium and the subsequent changes in its optical properties and motility can be monitored via changes of transmission and the resonance wavelength [201]. Furthermore, the metabolic rate in response to the antibiotic treatment can be determined by the alteration of the impedance of the medium surrounding the bacteria [201]. This method allows the simultaneous investigation of the electrical and optical response of an individual bacterium and it offers a rapid response [201].

Antimicrobial agents are becoming ineffective against many pathogenic bacteria, especially in the presence of biofilms [202]. Thus, there is a need for methods that can rapidly detect and treat biofilm infections as well [203]. Brunetti et al. proposed the design of a novel optoelectronic device based on a dual array of interdigitated micro- and nanoelectrodes [202]. This method is capable of monitoring the growth and maturation of the biofilm and probably testing the efficiency of antibiotics [202]. Subramanian and co-workers developed a threshold-activated feedback-based impedance sensor-treatment system for combined real-time detection and treatment of biofilms [203]. As a result, bioelectric effect treatment reduced the average biofilm surface coverage with ~74.8% compared to the untreated negative control [203]. In both cases, *E. coli* biofilm was used for the experiments. These techniques may be also used in the future for testing resistance, real-time detection of biofilms, and their in situ treatment, even on the surfaces of, for example, medical implants [203].

#### Bacteriophage in Biosensors

Bacteriophages are viruses of bacteria; they only infect bacteria and do not attack animal or human cells. Bacteriophages are ubiquitous in all natural environments and they are the most abundant living entities. They differ from each other in structure and physicochemical properties and are highly specific, recognizing only bacteria of one species or even only one particular strain [47,204]. Their recognition specificity is due to binding to a specific receptor on the bacterial surface. After recognition and binding, they inject their DNA into the bacterial cells. Bacteriophages are already being used in novel technologies to achieve fast recognition of pathogenic bacteria, particularly by using bacteriophage-based probes [47,205]. It is possible to isolate bacteriophages against any target bacterium. New real-time detection systems to recognize antibiotic-resistant bacteria would need to be developed as well; bacteriophages can be effectively used in different biosensors as recognition elements even for detection of these resistant pathogens [47]. Only a few commercial appliances detect antibiotic-resistant bacteria with bacteriophage-based technology (e.g., the blood culture test for MRSA/MSSA developed by MicroPhage Inc. [47] can be used as a clinical aid in combination with other laboratory tests to recognize MRSA from samples from patients; it is based on the bacteriophage amplification method and has a low detection limit (6 × 10^5^ CFU/mL) with waiting for 5 h to get the result). Commercial devices using bacteriophage amplification have been developed by BIOTEC Laboratories Ltd. for detection and testing of antibiotic resistance of *M. tuberculosis* in sputum [47,206,207]. Another US FDA-approved system is the xTAG Gastrointestinal pathogen panel (GPP) by Luminex Molecular Diagnostics [47]. The panel is for the identification and detection of multiple viral, parasitic, and bacterial nucleic acids in human stool samples from patients with symptoms of gastroenteritis or infectious colitis. Multiple pathogens, for instance *S. typhimurium*, are included in the. The method is based on the reverse transcription–polymerase chain reaction (RT-PCR/PCR) with a low detection limit of 2.3 × 10^5^ CFU/mL and a 24 h detection time in case of *S. typhimurium* [47]. A few of the antibiotic-resistant bacteria from the Centers for Disease Control (CDC) list have counterpart bacteriophages, but to our knowledge phage-based detection assays have not yet been developed [47]. In the work of Guntupalli et al., a penicillin-binding protein (PBP 2a) antibody-conjugated latex beads were utilized to create a biosensor to discriminate between methicillin-resistant (MRSA) and methicillin-sensitive (MSSA) *S. aureus* [208,209,210]. Lytic bacteriophages are formed into phage spheroids by contact with a water–chloroform interface [208]. When lytic phages are transformed into spheroids, they retain their strong lytic activity and show high bacterial binding ability. Phage spheroid monolayers were transferred to the surface of the biosensor by the Langmuir–Blodgett (LB) method where they were examined by a quartz crystal microbalance with dissipation tracking (QCM-D) to measure bacteria–phage binding [208]. Bacteria–spheroid binding results in reduced resonance frequency and a rise in dissipation energy for both MSSA and MRSA strains. The sensors are then exposed to the penicillin-binding protein–antibody latex beads after the bacterial binding. MRSA responds to PBP 2a antibody beads, but sensors with MSSA gave no response. Biosensors whose ability to detect resistant bacteria has been demonstrated are summarized in Table 2.

### 2.4. Strategies to Reduce and Control Bacterial Adhesion on Sensing Surfaces

One of the most decisive steps which precedes the harmful effect of bacterial infection on implants is microbial adhesion. It is very difficult to mechanically remove a matured biofilm or kill its bacteria [211]. Thus, inhibiting the bacterial surface-sensing mechanism and the subsequent initial binding events to surfaces is necessary to prevent biofilm-associated problems [211,212]. Efforts have been made to reduce adhesion using specific surface coatings (e.g., hydrophilic and antimicrobial coatings) [213,214]. Contradictory results from such treatments initiated a more theoretical approach in understanding the interactions involved in bacterium adhesion. A surface chemistry approach based on the DLVO theory describes two potential boundaries: the secondary and primary minima as a bacterium approaches a surface [215,216]. In this model, the bacterial cells are considered to be large particles and van der Waals and electrostatic interactions are taken into account. If the bacterium can surpass the energy barrier between the secondary and primary minimum (depending on the ionic strength), the cell can reach the surface (i.e., stays in the primary minimum). Although this simple model can explain irreversible adsorption as a function of ionic strength, the natural complexity of the adhesion cannot be covered and explained in full depth (Figure 2). Furthermore, at physiological ionic strengths the electrostatic interactions are negligible in comparison with electron donor–acceptor interactions [217]. In an extension to DLVO theory, van Oss et al. suggested an additional term, the Lewis acid–base (AB) interactions, which account for hydrogen bonding upon close approach of bacteria and substrate surfaces [218]. This XDLVO theory has been tested and improved over the years and is useful for predicting adhesion reversibility [219]. Nevertheless, surface chemical inhomogeneity and morphological roughness, and the living nature of the system, may also need to be taken into account [220], as was already demonstrated in the case of proteins [102,221].

It follows from the above discussion that screening the surface charge of a substratum directly changes the energy landscape of adhesion, provided ionic strength is low. Polymer-mediated interactions are not always repulsive, depending on the properties of the polymers and the solid surface [224]. The Whitesides group studied resistant surface coatings of various self-assembled monolayers (SAM) with different head groups and found that surfaces terminated with derivatives of Sarc and *N*-acetylpiperazine resisted the adhesion of *S. aureus* and *S. epidermidis* [225]. They used a commercial SPR biosensor to follow up the building of various SAMs on the gold sensor surface. These layers have been shown to resist protein adsorption in previous studies. The general hypothesis that protein-resistant layers also inhibit bacterial adhesion was examined. In summary, the authors found no correlation between the adsorption of protein (fibrinogen and lysozyme) and the adhesion of bacteria—unsurprisingly, since the outer surface of a bacterium is polysaccharidic rather than proteinaceous. It is a general belief that bacteria attach more rapidly to hydrophobic surfaces such as polystyrene and Teflon than to hydrophilic materials such as glass. However, results in the literature showed a more complex behavior and there are other important factors beside surface energy [226].

An important class of chemicals, namely the poly(ethylene glycol)s, has been used to modify surfaces with the aim of reducing bacterial attachment, based on this compound’s known high mobility and extremely large exclusion volumes [227]. It was found that poly(ethylene glycol) significantly reduced surface hydrophobicity and indeed reduced the attachment of *S. epidermidis* and *E. coli* by up to 2 log CFU/mL [228]. In another study, Roosjen and co-workers covalently coated glass surfaces with poly(ethylene oxide) brushes and demonstrated a reduction in the attachment of five bacterial and two yeast strains [229]. Covalent functionalization of glass with poly(4-vinyl-*N*-alkylpyridinium bromide) (a polycation material that has been reported to kill bacteria) can reduce the attachment of 1–2 log of *S. aureus*, *S. epidermidis*, *P. aeruginosa*, and *E. coli* on the surface [230]. In a seminal paper, Hook and co-workers applied a combinatorial chemistry approach for bacteria-resistant polymer surfaces [231]. They generated a large combinatorial space from a library of 22 acrylate monomers including ethylene glycol chains of various lengths, fluoro-substituted alkanes, linear and cyclic aliphatic, aromatic, and amine moieties. The combined monomer solutions were printed onto a poly (hydroxyl ethylmethacrylate) (pHEMA)-coated microscope slide to form a combinatorial polymer microarray. High-throughput surface characterization of microarrays was carried out to investigate the effect of polymer surface properties on bacterial attachment. The microarray was incubated with *P. aeruginosa*, *S. aureus*, and uropathogenic *E. coli* (UPEC) with plasmids expressing GFP (green fluorescent protein) and read by a fluorescent scanner. As an important result, hydrophobic moieties such as aromatic and aliphatic carbon groups when accompanied by the weakly polar ester groups were identified for the resistance of bacterial attachment. Surface sensitive label-free techniques such as OWLS (optical waveguide lightmode spectroscopy), RWG (resonant waveguide grating), and novel printing technologies (FluidFM BOT) are especially useful to quantify the repellent properties of thin films. For example, poly-l-lysine-grafted poly(ethylene glycol) copolymer coating reduced the adhesion of mammalian cells as well, not just bacterial cells as demonstrated by a RWG biosensor [232]. By applying FluidFM BOT technology, it is possible to print patterns from different materials to attract or to repel cells [233]. Furthermore, even coatings from bacterial proteins can be used as repellent surfaces. In the study of Kovacs et al., OWLS biosensor measurements revealed the bacteria-repellent properties of films fabricated from bacterial flagellin [1].

A different way to tackle the challenge of antimicrobial coatings is to take natural antibiotics, notably the antimicrobial peptides [234,235]. Synthetic antimicrobial peptides have been extensively studied recently [236]. These peptides were grafted to copolymers and were further modified by varying their hydrophobic groups to optimize their antibacterial and hemolytic activity [237].

The last group of antimicrobial surfaces we briefly summarize here is microbicide-releasing coatings. A well-known product for suppressing microbial growth is Microban. It incorporates triclosan (5-chloro-2-(2,4-dichlorophenoxy)-phenol), a broad spectrum phenolic antimicrobial, into a surface. The antibiotic then leaches from the surface to provide the bactericidal function. Another well-known example is silver, which has long been known and used; the metal ions (Ag^+^) have significant antimicrobial activity. There are more examples and recent results in the field of antimicrobial coatings, and the interested readers are advised to consult the literature [171,238].

It should be noted that next to development of bacteria-repellent surfaces, surfaces with adhesive properties are also important. Typically, a bacteria of interest can be effectively detected by employing specific antibodies. In a recent work by Farkas et al., different bacteria repellent (protein and synthetic polymer-based repellent coatings) and bacteria adhesive antibody-coated surfaces (using protein A, avidin-biotin, AnteoBind (previously called Mix&Go)-based surfaces) were developed and tested using OWLS [171]. The best setting in the biosensor measurements with *E. coli* was achieved by applying polyclonal antibodies in combination with protein A-based immobilization and PAcrAM-P blocking for nonspecific binding [171]. With these adjustments, 70 *E. coli* cells/mm^2^ surface sensitivity was demonstrated [171].

### 2.5. Biosensor Applications to Elucidate Molecular Modes of Action—Research Challenges

In order to fight bacterial resistance, it is equally important to develop novel antibiotics/antimicrobial agents and to identify agents that can attack new target structures or overcome the already existing resistance mechanisms. Biosensors can also be applied for searching for novel targets and simulated processes of antibiotic activities [21]. Mostly, SPR and mass-sensitive biosensors (such as the quartz crystal microbalance (QCM) or the surface acoustic wave sensor (SAW)) have been used to investigate the mode of action of established or experimental antibiotic compounds and other antibacterial candidates. In general, the classical microbiological assays determine the MIC values (minimal inhibitory concentration) and bacterial killing kinetics, which constitute the selection criteria for further development. Biosensors are useful for demonstrating an interaction or binding of antibiotics to bacterial components, despite the fact that a quantitative and physico-chemically rigorous interpretation of the data is rarely attempted and, given the dubious assumptions and lack of complete data, is often unattainable. It can be also interesting in basic research to know the exact binding parameters, for example, the Kd value (dissociation constant), between the bacteria and the examined surface/specific antibodies, or even the binding between their adhesive molecules and other ligands.

The adhesive molecules of bacteria which can be studied in the future more deeply (i.e., their roles in attachment to different surfaces or in other reactions, cell entry, etc.) are summarized in the next section.

## 3. Biological Mechanisms in Bacterial Adhesion as Potential Subjects of Label-Free Monitoring

In this section, we provide a brief summary of bacterial adhesion molecules. The process of bacterial cell entry into host cells and the role of glycocalyx is also mentioned to provide a whole picture of the bacterial adhesion which can be further explored with novel, sensitive label-free biosensor technologies. These appliances provide kinetic data of the processes, and even new antibacterial agents or other compounds can also be easily tested in a real-time and fast way, in a high-throughput manner to understand the efficacy and mode of action of these chemicals on bacteria.

### 3.1. Adhesive Molecules of Bacteria and Bacterial Entry into Host Cells

Bacteria have extensive tools—from single monomeric proteins to multimeric macromolecules—used for adhesion and invasion [239]. The infection starts with the adhesion of a bacterial pathogen to the host cell’s surface. In general, pathogenic bacteria express surface-associated proteins, called adhesins, that recognize components of the eukaryotic cell surface or ECM and thus enable close contact with the host tissues [17,240]. Bacterial adhesins may act synergistically, thus increasing their adhesive capacity [241]. Over the course of infection bacteria may express different adhesin types by gene regulation, depending on the site of infection, various environmental [242] and mechanical signals [243], and quorum sensing [244]. This gene regulation enables the bacteria to successfully colonize the host, first switching from reversible to irreversible adhesion, and later allows biofilm formation [245,246]. Alterations in metabolic pathways (for example, induction of stress responses) usually affect biofilm formation and cell motility; furthermore, bacterial phenotype can also change during biofilm maturation [247].

Adhesins mediate attachment of pathogenic bacteria to host cells. These molecules are considered to be important virulence factors, but not all adhesins can be considered as essential virulence factors. Some pathogens produce more than just one adhesin, which can be expressed at different locations and times in the host and contributes individually to host cell attachment [17]. A variety of adhesins may also promote/initiate cell invasion [248]. Adhesins can be divided into two main groups: fimbrial and afimbrial [17,249].

Fimbrial adhesins are filamentous polymeric structures in Gram-negative bacteria [241]. Initial adherence of microbes usually occurs via these structures (called fimbriae or pili), which extrude outwards from the bacterial cell surface [17,250]. The formation of pili is complex and involves several auxiliary gene products that transport the pilus subunits to the cell surface and compose them into the organelle. The first step in their creation is the secretion of the pilin and adhesin subunits to the periplasmic space [17]. Host cell receptor proteins of pili are generally carbohydrate residues of glycolipids or glycoproteins [251]. For instance, in the upper urinary tract, P pili bind to the α-d-galactopyranosyl-(1–4)-β-d-galactopyranoside moiety of glycolipids of cells [252]. Some bacteria, including Salmonella and Neisseria, have special environmentally controlled mechanisms to switch between nonpilated or pilated phases or different pilus types [17,249,253,254]. Some bacteria use their pili as invasins [17,252,255].

The group of afimbrial adhesins contains all surface proteins that are necessary for adherence but do not form themselves into pili-like structures [256]. According to some researchers, they are necessary to mediate a closer intimate attachment to cells that follows initial connection via fimbriae [242,257]. These proteins usually contain parts that are homologous among each other or are homologous to eukaryotic sequences that enable cell-to-cell adhesion [17]. Afimbrial adhesins of *E. coli* (intimin), *Neisseria* (Opas), *Yersinia* (invasin, YadA), *Staphylococcus*/*Streptococcus* (fibronectin-binding proteins, FnBPs), and *Listeria* (internalin A, B, InlA, InlB) are the most characterized.

*E. coli* strains (enteropathogenic and enterohemorrhagic) produce the outer membrane protein intimin that recognizes minimum two types of receptors on the surface eukaryotic cell [17]. The first type includes those recognizing the β1 integrins, members of a big family of eukaryotic ECM receptor molecules [243]. The second type recognizes the translocated intimin receptor (Tir), which is produced by the bacteria and inserted into the host membrane [244]. The intimin molecules of different *E. coli* strains are about 900 amino acids in length and have an N-terminal region that is very homologous to invasin. The cell-attaching function of invasin and intimin is localized within the last C-terminal 200 amino acids. Fibronectin, a substrate of β1 integrins, also contains two aspartate residues necessary for receptor binding. The similarity within the C-termini of intimin and invasin is the presence of a pair of cysteine residues separated by approximately 70 amino acids [17]. These residues form a disulfide loop in the invasin and intimin that is necessary for adhesion to β1 integrins [28].

*Neisseria meningitidis* and *Neisseria gonorrhoeae* express a number of Opas and Opcs, which are highly variable opacity-associated outer membrane proteins [258,259]. These proteins impart tight interactions between various eukaryotic cells [245,246]. A minimum of 12 diverse variants of Opa genes are in the gonococcal genome and they are expressed independently of each other [245,246]. The Opa genes are transcripted from their own promoter [260]. Switching in Opa expression may contribute to antigen variability and various receptor tropisms [17,249]. For example, the Opa50 variant recognizes heparan sulfate proteoglycan (HSPG), fibronectin, and vitronectin, which plays a role in the attachment of the bacterium to β1 integrin receptors [261]. Both β1 integrins and HSPG and are on the basolateral side of epithelial cells and primarily contribute to the basolateral invasion of epithelial cells [262]. Opa- and Opc-mediated cell invasion and interaction are inhibited when lipooligosaccharides are modulated by sialylation [17,263].

Enteropathogenic *Yersinia* has an additional adhesin, termed YadA, that helps adhesion to epithelial cells [264]. YadA also defends the bacterium against the defensin lysis and complement system [264]. YadA is a member of the family of adhesins that form lollipop-shaped surface projections (for example, UspA1 and UspA2 of *Moxarella catarrhalis*), and encoded on the *Yersinia* virulence plasmid. Three or four YadAs form a fibrillar structure and cover the surface of the bacterium as a capsule-like fibrillous matrix. The C-termini of the YadAs are responsible for oligomerization, translocation, and fixing in the outer membrane [17]. The N-termini fold into the global head domain that plays a role in adhesion to cells and ECM molecules (collagens, fibronectin, laminins) [265]. It has been demonstrated that the eight repeats of the AsnSerValAlaIleGlyXxxSer amino acids in the N-terminal part of YadA are principally involved in collagen binding. YadA also promotes the penetration into host cells via β1 integrins [266].

*L. monocytogenes* is a foodborne pathogen that penetrates into different cell types during disease. This bacterium synthesizes various homologous surface proteins, termed internalins, which are significant for bacterial entry into eukaryotic cells [17,267]. Internalins bind to ECM molecules with high affinity and, in general, the binding is irreversible. They are covalently linked to the cell wall at their C-terminus, and the N-terminus is exposed to the environment. Mutants are attenuated in virulence, which indicates that these molecules participate in colonization and disease as well [17]. Mengaud and co-workers [268] have identified the corresponding receptor as E-cadherin, a transmembrane surface molecule of mammalian cells which plays a role in the formation of homophilic cell–cell junctions [17]. The InlA-E-cadherin-promoted penetration of Listeria corresponds to the invasin-1β-integrin-mediated cell entry by *Yersinia* [269]. Both bacteria penetrate via the ‘zippering’ mechanism [270] (see Figure 3). A schematic illustration of the parts of a bacterium and the adhesive molecules is found in Figure 1.

#### 3.1.1. Type III Secretion-Associated Invasion Mechanisms

*Salmonella* and *Shigella* use a trigger mechanism to induce their penetration into host cells (see Figure 2) [271]. The constituents that mediate invasion in both microbes are homologous and involve a type III secretion system which uses effector proteins into host organisms [272]. This induces host-signaling pathways required for cytoskeletal rearrangements resulting in bacterial penetration [17]. Type III secretion systems are found in plant, animal, and human pathogenic bacteria and contain approximately 20 proteins that are very homologous between the various species [273]. In general, type III secretion proteins are associated with the bacterial membrane, forming a secretion pore complex with a short hair-like structure that resembles an injection needle, like the flagella motor and the hook [17,274]. The induction of the type III secretion system involves the mobilization and association of proteins which already exist in the bacterial cytosol [17]. A small number of invasion factors can be competent to bind to the receptors of the host organism, which then induce the signal for the uptake of the pathogens [275]. Secreted Ipa proteins in *Shigella* [276] and Sip proteins in Salmonella are required for invasion [277]. *Shigella* Ipa proteins can interact with α5β1 integrins, which lead to the activation/ phosphorylation of the pFAK125 (focal adhesion kinase) and paxillin (cytoplasmic adapter protein) [17,278,279]. IpaB binds to the extracellular domain of the receptor CD44 [280]. CD44 cell-surface glycoprotein is recruited at bacterial entry sites and appears to be necessary for Shigella invasion by starting the early steps of penetration [280].

#### 3.1.2. Receptors on the Cell Surface

The host receptor that is recognized by the adhesin of the bacteria defines the tissue and organ specificity, as well as the colonization and/or entry site [281,282]. Various hosts and tissues vary in the distribution and amount of the receptors, and the differential expression of the adhesin/invasin can further intensify the specificity for certain host cell types [281].

#### 3.1.3. Adherence to ECM Proteins

Many adhesins/invasins of different pathogenic bacteria have been characterized—they recognize certain ECM proteins [283]. Some of them can bind to many ECM components and to cell receptor molecules as well, for example integrins [284]. In general, the bacterium does not distinguish between immobilized, tissue-specific, and soluble derivatives of ECM proteins [285]. Pathogenic bacteria interact specifically with a certain form of a matrix protein via several various strategies. Fibronectin is present in many secretions, most wounds and fluids [17]. Thus, it is understandable that wound- and mucous surface-colonizing microbes, for instance *Streptococcus* and *Staphylococcus*, evolved special fibronectin-binding proteins (FnBPs) as adhesive factors [286]. In general, ECM components themselves interact with a wide range of cell receptors [287].

#### 3.1.4. Specific Cell Receptors Involved in Cell–Cell Interaction

A lot of adhesive factors of pathogenic bacteria recognize transmembrane cell receptors that may contribute to cell adhesion, for example, cell–cell adhesion junctions or focal contacts [17,288]. These receptor molecules fall into four groups: the integrins, selectins, cadherins, and the immunoglobulin superfamily [289]. They have roles in mediating contact to the cytoskeleton, in the assembly of extracellular adhesion sites, and in inducing signal transduction cascades exerting major short- and long-term responses [290]. A broad range of pathogenic bacteria with various phylogenic origins bind to integrin receptors that are especially involved in cell–matrix and cell–cell interactions and are present on many cell types [291]. The integrin family comprises more than 20 transmembrane proteins. These members form a noncovalently associated heterodimer with an α and a β subunit; 16 various α and 8 various β variants. The extracellular regions of a particular α and β1 chain form a specific extracellular globular head, with a ligand binding pocket that determines the substrate specificity of cells and recognizes distinct ECM ligands. The α chain has multiple repeats of divalent cation-binding sites called EF-hand motif that mediate the regulation of ligand binding [17]. Generally, β1 integrin ligands have a sequence motif Arg-Gly-Asp (RGD) which plays an important role in integrin interaction [17,292]. Amino acid Asp or Glu seems to be directly involved in receptor binding [293]. Many pathogens penetrate into mammalian host cells via β1 integrins [294]. Invasin-mediated cell entry of *Yersinia* by β1 integrins is very effective [295]. In a study by Krukonis and Isberg [296], the authors showed that natural ECM substrates and invasin recognize the same domain in the α and β1 integrin chain and apply similar structural determinants. The bacterial invasin binds to integrins with greater affinity than fibronectin, a host integrin ligand [17,297]. The activity of cytoskeletal proteins and signaling are ruled by the multimeric state of the integrin receptor [17,298]. As invasin-promoted bacterial penetration is intensified by multimerization, it has also been speculated that invasin multimers with multiple receptor binding sites interfere with several β1 integrin molecules. Yersinia’s YadA protein that plays role in invasion via β1 integrins also creates multimers in the outer membrane of bacteria [17]. Hence, these multiple ligand-receptor interactions may fixate adhesion and intensify the uptake procedure through integrins [295].

β2 integrins of leukocytes also present an important entry point for pathogenic bacteria [17]. The CR3 complement receptor, the αmacβ2 integrin binds complement component C3b and mediates the entry of *Yersinia* [299], *Mycobacterium* [300], *Legionella* [301], and *Bordetella* [302]. Bacterial entry αmacβ2 integrin is performed indirectly by the deposition of complement factor C3b or by direct interaction of an adhesin with the receptor. In general, bacteria are killed in the lysosomes; however, *Legionella* and *Mycobacterium* survive and grow in phagocytes. Binding to integrin receptors is an essential strategy of pathogenic bacteria to interact with the cytoskeleton, trigger adhesion and penetration in a broad range of various cell types. Cadherins support Ca^2+^-dependent cell–cell interaction [17]. The sequence His-Ala-Val (HAV) in the first module is fixed among cadherins and is necessary for homophilic binding [17,303]. Many more cell surface molecules have been identified that play roles in the adhesion and invasion procedures of bacteria [304].

#### 3.1.5. Bacterial Entry into Host Cells

The lifestyle inside the host cell has certain advantages to the pathogen bacteria; they no longer have to undertake energy-consuming adherence procedures to remain at the site of infection, they are further defended from the host immune system and antibiotics, and a wide range of nutrients are available in the cell cytoplasm [305]. Invasion offers the probability to penetrate and spread into deeper tissues and thus constitutes one of the most important strategies for bacteria to remain alive and be disseminated in their host organism [306]. In addition, intracellular pathogens need some virulence determinants, not just to pass into the cell, but also to withstand intracellular circumstances that may inhibit intracellular growth [17,307,308]. The host cell is not passive in the uptake process; moreover, pathogens usually apply special invasion mechanisms that exploit normal signaling pathways for their own benefit [306]. Although each species has adopted certain mechanisms to penetrate into eukaryotic cells, they generally share common elements, such as receptors (adhesion molecules), cytoskeleton-associated proteins, and signaling molecules (tyrosine kinases, GTPases, PI3-kinase) to confuse cellular functions [309].

During infection, phagocytosis is the main mechanism of the host to remove pathogenic microbes, by, for example, monocytes, neutrophils, and macrophages (see Figure 4) [17]. Phagocytosis is triggered by the cross-linking of various phagocytic receptors, resulting in the remodeling of the actin cytoskeleton and the cell membrane leading to the engulfment of the bacterium and formation of the phagosome, where eventually, through phagosome maturation, most bacteria will be killed and digested [310]. However, some bacteria can exploit this mechanism, surviving and proliferating in the host’s professional phagocytes; *Mycobacterium tuberculosis*, for example, may inhibit phagosome maturation by preventing its acidification while *Legionella pneumoniae* blocks the fusion of the phagosome with the lysosome, among multiple other strategies [311].

*Salmonella* and *Shigella* use the triggering mechanism to penetrate into host cells, where a brief contact with the surface of the host cell actuates a fast cytoskeletal alteration in which actin polymerization in the cytoplasm pushes out large membrane extensions, called membrane ruffles (see Figure 4) [312,313]. This Type III secretion-associated invasion mechanism leads to macropinocytosis, when the extended ruffles fold over and fuse back to the cell [17]. This procedure results in the inclusion of huge membrane pockets with the microorganism next to the in-folding ruffles [17]. This kind of invasion is very fast; the pathogen appears in the membrane-bound vacuole in a few minutes after contact with the host cell [314]. Salmonella triggers activation of host cell’s phospholipase C, which provokes a temporary increase in the inositol triphosphate (IP3) concentration and mobilizes Ca^2+^ from intracellular stores [272]. Two actin binding proteins of Salmonella play a role in cytoskeletal redistribution; SipC is necessary for F-actin bundling and nucleation of actin polymerization during the entry, whilst SipA protein stimulates the function of SipC [17,315].

The zipper-type invasion procedure includes a single bacterial surface ligand that tightly contacts host cell adhesion molecules (see Figure 4) [17]. Thus, the entry process is driven by the tight binding between bacterial adhesins and their targets [270]. This connection leads to the clustering of receptors at the bacterium–cell interface that induces signal transduction cascades with cytoskeletal alterations (pseudopod-like structures). Consequently, circumferential binding of the receptors about the surface of the pathogen elicits pseudopod-like structures surrounding the microbe and eventually results in the penetration of the bacterium into the so-called bacterial phagosome, a membrane-bound vacuole. The cadherin and integrin receptors that play a role in the internalin-mediated entry process are connected with the actin cytoskeleton via a multifactoral protein complex that links the actin filaments with the cytoplasmic domains of the receptors [17]. This method is used by *Listeria* and *Yersinia*, Gram-positive and Gram-negative bacteria, applying a different host cell receptor [17,316].

Figure 4 shows the possible ways of bacterial entry into host cells.

### 3.2. Role of Bacteria’s Glycocalyx in Their Adhesion

Bacterial glycocalyx is a highly hydrated polymeric layer that extends outwards from bacterial surface; 99% of its volume is water [19]. Branching sugar molecules or a mass of tangled fibers of substituted polysaccharides form the glycocalyx that surrounds an individual cell or colony [18,317]. Bacteria must expend energy to generate and maintain a glycocalyx. Thus, in the environment of a pure and unthreatened culture, it is an unnecessary material conferring no selective advantage; cells that create these coatings are usually eliminated from pure cultures by uncoated ones that can allocate more energy to proliferation. In natural, competitive circumstances, selection prefer cells that are protected and able to adhere to a proper surface by their glycocalyx. Thus, having a glycocalyx is required for most bacteria to succeed in their different natural environments [18]. It can be visualized by ruthenium red staining, which is taken up by any polysaccharide present in the glycocalyx fibers [318].

The constituents of the glycocalyx may be either homopolymers or complex heteropolymers of monosaccharides (mainly neutral hexoses, 6-deoxyhexoses, polyols, uronic acids, and amino sugars), with substituents formate, pyruvate, phosphate, and succinate.

Glycocalyxes are divided into two types based on how tightly they are attached to the bacterial cell. Loosely attached, diffuse fibers form the amorphous slime layer, while in the capsule, the bacteria is tightly covered with a gelatinous, distinct fibrous matrix, in which the forming polysaccharide is covalently attached [319].

Physiological factors regulate the synthesis of the glycocalyxes [247,320]. Mostly, capsule formation is favored in case of high C:N ratio of the environment, and ion concentrations have been shown to affect capsule production as well [19,321]. When the glycocalyx is lost (in laboratory circumstances), other polymers are presented (for example, the LPS of the Gram-negative bacterial cell wall) [19]. However, during infections, bacteria may also change their expression of antigenic determinants, including polysaccharides, in a process called phase variation, which may well help them evade the host immune response and thus affect their virulence [322]. The highly variable nature of bacterial polysaccharide expression is also reflected in the relatively high number of serotypes identified in certain bacteria [319,323]. Importantly, some bacteria even express polysaccharides mimicking host molecules to avoid recognition by the immune system [324].

Bacterial adhesion to an inert surface must overcome the electrostatic repulsive forces [325]. Long hydrophilic extensions of the bacterial cell can bridge the space between surface and bacterium and allow H-bonding or ion pair formation (attractive forces) to start attachment [19,326]. After its adhesion to the surface, the cell surrounds itself with further glycocalyx fibers and multiplies itself within this matrix to form a microcolony [18,19]. Other morphologically distinct bacteria then adhere by their glycocalyx to create a layered bacterial consortium, forming a biofilm [16,19,317,327,328].

The negatively charged glycocalyx can form a polar bond with polysaccharides via divalent positive ions in the medium, and lectins can also form a bridge between them. The nature of the host/eukaryote glycocalyx changes as the cell ages. Furthermore, the glycocalyx alters in the cells that have been infected by a virus, which may explain the enhancement in susceptibility to bacterial illness that is often observed after viral infection [18]; viral infection changes the tissue cell surface and allows the colonization by opportunistic bacteria [19]. Adhesion to a certain tissue provides bacteria a constantly renewed supply of organic nutrients with physical conditions favoring growth. Because of adhesion, infecting bacteria can be resistant to removal, especially in a system that is sterile among normal conditions. Pathogens in the urinary tract have well-developed glycocalyxes that protect the cells from being washed out with the urine. The specificity of some bacteria that invade only a certain host tissue is in correlation with the specificity of the glycocalyx for the host-tissue cells [18].

In contrast, pathogenic bacteria adhere to tissues and induce acute disease by the production of enzymes/toxins or adhere to a tissue and multiply until they inhibit organ function and cause a cryptic disease [19]. Animal pathogens within infected tissues have thick glycocalyxes [19,329].

Most importantly, the glycocalyx prevents desiccation of the bacteria and also plays a role in retaining digestive enzymes produced by the bacterial cell and directing them against the host cell. Similarly nutrient molecules and ions in the immediate environment can be bound in the glycocalyx. In some illnesses, the production of bacterial glycocalyxes and thereby the virulence of bacteria seems to be increased by increased concentrations of particular nutrients. Attachment to a surface protects the bacterial cells from particular protozoans, furthermore, the glycocalyx is a physical margin against bacterial viruses and predatory bacteria [18]. The glycocalyx prevents host antibodies binding to bacterial antigens, for example, in case of *P. aeruginosa* in the urinary tract [18,319].

In many chronic illnesses, bacteria form biofilms, which are communities of bacteria encased in their glycocalyx on different surfaces [18,317,330,331,332,333]. The general stages of biofilm development are the adhesion of planktonic bacteria to the surface (reversible-irreversible attachment, influenced by hydrodynamic forces, gravitational forces, nutrient levels, chemotaxis, ionic strength, pH, temperature, and adhesive molecules), microcolony formation, biofilm maturation (gene expression alterations, upregulating factors favoring sessility, increase of eDNA amounts, regulation of DNA release by Atn autolysin) and dispersal (escape from the matrix) [334]. Bacteria in biofilms are very resistant; they can be 1000 times more resistant to antimicrobial treatments than planktonic bacteria of the same species [330,331]. Biofilm formation is the serious clinical problem—65% of all human bacterial infections involve biofilms [330,333]. Due to their high resistance to antibiotic treatments, the best treatment for foreign body-associated biofilm infections is to remove the infected device, but the removal is difficult in some cases such as prostheses, cardiac implants, and pacemakers [334,335]. Therefore, numerous studies have focused on ablating bacterial adherence as the first step of biofilm formation with mannosides, pilicides, and curlicides [334]. Other agents inhibit the enzymes involved in the synthesis or modification of matrix components destabilize biofilm architecture by interfering with bacterial membrane stability or alter signaling cascades [334].

We would like to point out that the term “glycocalyx” was abandoned in favor of “extracellular polymeric substances” (EPS) [336]. In 1978, the matrix material was considered to be polysaccharides [18,336]. Later, it turned out that lipids, nucleic acids, and proteins are its other major constituents [336]. The role of the EPS matrix turns out to be fundamental for maturing biofilms [336].

## 4. Conclusions

We have reviewed the status of label-free biosensors in bacteria monitoring and summarized potential novel application directions with biological relevancies. Optical, mechanical, and electrical sensing technologies were all discussed with their detailed capabilities in monitoring bacterial species. In order to review interesting potential future applications of the outlined techniques in bacteria research, we summarized the most important kinetic processes relevant to the adhesion and survival of bacterial cells. These processes are potential targets of kinetic investigations employing modern label-free technologies in order to reveal new fundamental aspects and to initiate new directions for future research and development. As a potential future area to be targeted by these systems, we have outlined how microscopic pathogenic creatures enter much bigger mammalian cells and infect them. Not less importantly, we highlighted that numerous illnesses caused by pathogenic bacteria can be cured by antibiotics or other types of antimicrobial compounds, but excessive and sometimes incorrect usage has led to the proliferation of (multi)resistant bacteria, causing a worldwide problem. Pathogenic bacteria and antibacterial agents are important targets for detection—i.e., both cells and molecules. This review, however, focuses above all on the detection of bacterial cells (the detection of eukaryotic cells has been reviewed elsewhere [170]). The use of biosensors to detect molecules is covered elsewhere [337,338]. We summarized the roles of adhesive molecules, receptors, and glycocalyx of the bacteria, which are highly important factors in biofilm formation. To reduce and inhibit biofilm formation and maturation, bacteria-repellent surfaces are being developed and tested. For a long time, culture-based conventional methods were the only way to identify bacteria from samples from, for instance, the food industry or medicine. Although they are reliable methods, they have long processing times and professional operators are required. In contrast, biosensors provide a rapid, simpler way to detect and identify bacteria and antibiotic effects, and modern commercial instruments can be used as a “black box” by untrained staff ignorant of their inner workings, often with acceptably reliable results. We have reviewed the most promising types of label-free biosensors, including those that use bacteriophages. Recently, the examination of antibiotic resistance by using these techniques has been implemented. Further improvements of biosensors should make the detection of pathogens easier and cheaper, and they may become used routinely in clinical medicine, the food industry, security, and public health more effectively [339]. Moreover, thanks to their unique capabilities, novel label-free sensing technologies will potentially find applications in basic research along the lines summarized here.

## Figures and Tables

**Figure 1 biosensors-12-00188-f001:**
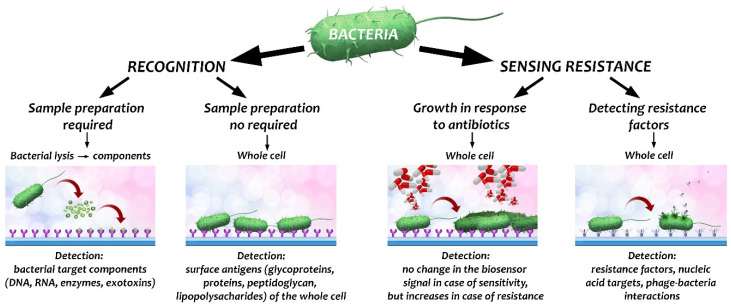
Summary of the strategies of recognition of bacteria and ways of resistance sensing using biosensors. Sample preparation may be needed to lyse the bacteria (or otherwise disrupt them) to liberate the target bacterial components (first column); and preparation-free whole cell-based assays are in the second column. Few biosensors can sense antibiotic resistance as well. There are two possibilities: measuring and monitoring the growth of bacteria during antibiotic treatment (third column) or measuring resistance factor adhesion or bacteriophage–bacterium interaction.

**Figure 2 biosensors-12-00188-f002:**
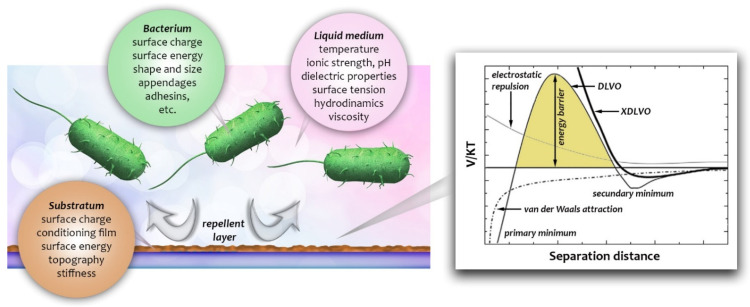
On the left: schematic illustration and the list of the parameters that affect the initial attachment of bacterial cells to a solid–liquid interface (in case of the bacterium, liquid medium and substratum [222,223]). On the right: plotted DLVO and extended DLVO (XDLVO) force graphs based on the figure of Hotze et al. [216], van der Waals force (dashed line), and electrostatic repulsion curves are also shown. V/KT represents the potential energy (V) divided by Boltzmann’s constant (K) and absolute temperature (T) [216].

**Figure 3 biosensors-12-00188-f003:**
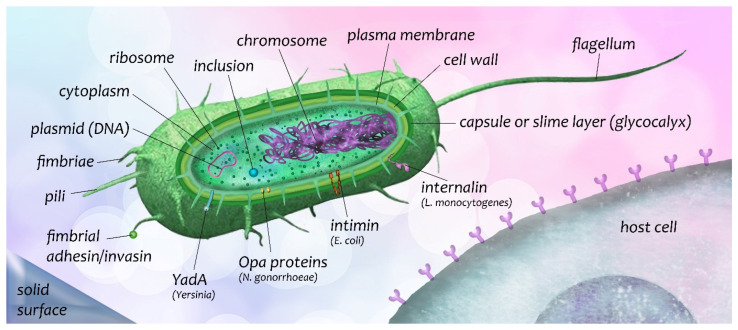
Schematic illustration of bacterial cell structure. The main types of adhesive molecules are also illustrated. Adhesins mediate attachment of pathogenic bacteria to host cells. Adhesive molecules that also promote the internalization are called invasins. However, not all the bacterial species have all the illustrated adhesive molecules. For example, enteropathogenic *Yersinia* has YadA, that helps to adhere to epithelial cells. *N. meningitidis* and *N. gonorrhoeae* express a number of Opas and also Opcs, highly variable opacity-associated outer membrane proteins. *E. coli* strains (enteropathogenic and enterohemorrhagic) produce the outer membrane protein intimin that recognizes minimum two types of receptors on the surface eukaryotic cell. *L. monocytogenes* synthetizes various homologous surface proteins, termed internalins, which are significant for bacterial entry into eukaryotic cells. Furthermore, bacteria can adhere to a plant, animal, or another bacterial cell by juxtaposing its own glycocalyx to the surface of the desired cell. Bacterial adhesion to an inert, solid surface must overcome the evolving electrostatic repulsive forces.

**Figure 4 biosensors-12-00188-f004:**
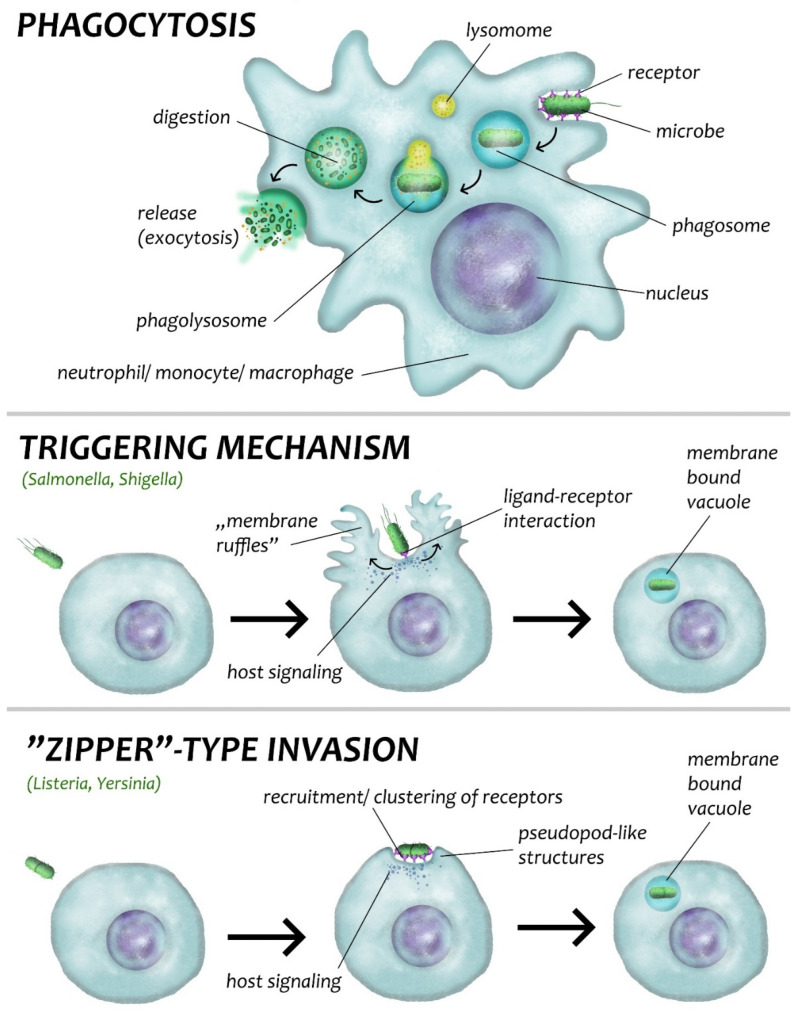
Schematic illustration of the ways of bacterial entry into host cells. In general, phagocytosis is the route of invasion for bacteria, the accession into professional phagocytes, for instance monocytes, neutrophils, and macrophages [15]. *Salmonella* and *Shigella* use the “triggering” mechanism their penetration into host cells, where a brief contact with the surface of the host cell eventuates a fast cytoskeletal alteration in which explosive actin polymerization in the cytoplasm pushes out large membrane extensions, the membrane ruffles [15]. The “zipper”-type invasion procedure includes a single bacterial surface ligand that tightly contacts host cell adhesion molecules. Thus, the entry process is driven by the tight binding between cell adhesion molecules and bacterial adhesins [15]. Adapted with permission from ref. [15]. Adapted with permission from ref. [15].

**Table 1 biosensors-12-00188-t001:** Summary of the whole bacteria detecting biosensors. In case of SPR and QCM, the different sensing strategies result in various limit of detection values (LOD). Thus, according to these results, the sensitivity of the biosensors can be improved by some development, for example applying nanoparticles. In general, the assay time is less than 1 h. (* represents label-based biosensors for direct comparison with label-free techniques).

Biosensor Type	Detected Bacteria	Limit of Detection	Assay Time	Ref.
SPR (with nonspecific adsorption of antibody)	*E. coli*	4.8 × 10^5^ CFU/mL	No data	Torun et al. [78]
SPR (with specific adsorption via avidin-biotin interaction)	*E. coli*	6.2 × 10^3^ CFU/mL	No data	Torun et al. [78]
SPR (with SAM formation of antibody)	*E. coli*	35 CFU/mL	No data	Torun et al. [78]
SPR (with gold coated magnetic nanoparticles) *	*E. coli*	3 CFU/mL	<70 min	Torun et al. [78]
SPR (with one- and two step sandwich assay)	*S. typhimurium*	4.7 × 10^5^ CFU/mL (one-step sandwich assay) 1.6–1.9 × 10^6^ CFU/mL (two-step sandwich assay)	No data	Bhandari et al. [70]
SPR (full-length Det7 bacteriophage tail protein (Det7T))	*S. typhimurium*	5 × 10^4^–5 × 10^7^ CFU/mL	∼20 min	Hyeon et al. [76]
BaTiO3-graphene-affinity layer–based SPR	*Pseudomonas*	No data	No data	Mudgal et al. [80]
Resolution-optimized prism- based SPR imaging (RO-SPRI)	*L. monocytogenes* and *L. innocua*	2 × 10^2^ CFU/mL	7 h	Boulade et al. [137]
Localized surface plasmon resonance (LSPR)	*V. cholerae O1*	10 CFU/mL	1 h	Faridfar et al. [138]
Localized surface plasmon resonance (LSPR)	*E. coli O157:H7*	10 CFU/mL	2 h	Yaghubi et al. [139]
Fiber-optic SPR	*E. coli O157:H7*	5 × 10^2^ CFU/mL	No data	Zhou et al. [68]
Fiber-optic LSPR	*S. typhimurium*	128 CFU/mL	100 min	Xu et al. [81]
SPR	*E. coli*, *Salmonella* spp.	17–57 CFU/mL for *E. coli* and 7.4 × 10^3^–11.7 × 10^3^ CFU/mL for *Salmonella* sp.	<80 min	Vaisocherová-Lísalová et al. [75]
SPR	*E. faecalis*	3.4 × 10^4^ CFU/mL	No data	Saylan et al. [67]
SPR	*M. tuberculosis*	10^4^ CFU/mL	15 min	Trzaskowski et al. [74]
SPR	*Pseudomonas* and *Pseudomonas*-like bacteria	no data	No data	Kushwaha et al. [140]
SPR	*Pseudomonas*-like bacteria	no data	No data	Maurya et al. [79]
SPR	*E. coli*	3.0 × 10^2^ CFU/mL	No data	Galvan et al. [141]
SPR	*M. tuberculosis*	63 pg/mL	2–4 h	Prabowo et al. [142]
SPR	*V. cholerae*	50 CFU/mL	60 min	Taheri et al. [71]
SPR	*E.coli O157:H7*, *S. enteritidis*, and *L. monocytogenes*	14, 6, and 28 CFU/25 g (mL)	No data	Zhang et al. [72]
SPR	*E. coli K-12*	2 × 10^4^ CFU/mL	20 min	Shin et al. [77]
SPR	*A. salmonicida*, *A. hydrophila*, *V. harveyi*	no data	No data	Padra et al. [69]
SPR	*S. typhimurium*	10^5^ CFU/mL	10 min	Makhneva et al. [73]
SPR	*L. monocytogenes*	3.25 log CFU/100 μl	7.5 min	Raghu et al. [66]
SERS	*Various bacteria species*	~single bacterium	No data	Yu et al. [87]
SERS	*E. coli*, *S. typhimurium*, and *B. subtilis*	no data	No data	Prakash et al. [143]
SERS	*E. coli O157:H7*	3 CFU/mL	No data	Zhou et al. [86]
SERS	*S. aureus*	1–1× 10^6^ CFU/mL	30 min	Lee et al. [88]
SERS	Methicillin-resistant *S. aureus* (MRSA), methicillin-sensitive *S. aureus* (MSSA), *S. aureus 29213*, *S. aureus 25923*, *C. albicans*, *B. cereus*, *E. coli*, *P. aeruginosa*	1.0 × 10^8^ cells/mL (used concentrations)	45 min	Chen et al. [144]
OWLS	*E. coli* *L. plantarum*	No data, purpose of work was to determine kinetics	2 min	Yeh et al. [145]
OWLS	*E. coli BL21AI* *E. coli B200*	2 × 10^9^ CFU/mL was applied	~150 min	Adányi et al. [146]
EC-OWLS	*L. plantarum 2142*	10^2^–10^3^ CFU/mL	~115 min	Adányi et al. [147]
OWLS (reverse symmetry waveguide design using nanoporous substrate)	*E. coli K12*	60 cells/mm^2^	Minutes	Horvath et al. [90]
MCLW	*B. subtilis var. niger bacterial spore*	8 × 10^4^ spores/mL	60 min	Zourob et al. [99]
MCLW with ultrasound standing waves	*B. subtilis var. niger*	1 × 10^3^ spores/mL	3 min	Zourob et al. [98]
MCLW with an electric field	*B. subtilis var. niger*	1 × 10^3^ spores/mL	3 min	Zourob et al. [96]
QCM (with direct binding assay)	*S. typhimurium*	~2 × 10^2^ CFU/mL	5 min	Salam et al. [113]
QCM (with sandwich assay)	*S. typhimurium*	~1.01 × 10^2^ CFU/mL	9 min	Salam et al. [113]
QCM (with nanoparticle amplification)	*S. typhimurium*	10–20 CFU/mL	12 min	Salam et al. [113]
QCM	*S. aureus*	5.18 × 10^8^ CFU/mL	No data	Pohanka [148]
QCM	*B. subtilis*	no data	No data	Latif et al. [111]
QCM	*C. jejuni*	150 CFU/ mL	No data	Masdor et al. [108]
QCM	*E. coli O157:H7*	1.46 × 10^3^ CFU/mL	50 min	Yu et al. [104]
QCM	*S. typhimurium*	10^3^ CFU/mL	1 h	Wang et al. [105]
QCM	*S. typhimurium*	10^0^ CFU/mL	No data	Fulgione et al. [109]
QCM	*S. typhimurium*	10^5^ CFU/mL	10 min	Makhneva et al. [73]
QCM	*E. canis*	4.1 × 10^9^ molecules/μL of 289 bp *E. canis*	No data	Bunroddith et al. [107]
QCM	*B. cereus*	no data	10 min	Spieker et al. [112]
QCM	*A. hydrophila*	1.25 × 10^7^ CFU/mL	5 min	Hong et al. [110]
QCM	*B. melitensis*	1.0^2^–1.0^7^ CFU/mL	No data	Bayramoglu et al. [106]
Asymmetrically anchored PEMC (aPEMC)	*L. monocytogenes*	10^2^ cells/mL	~30 min	Sharma et al. [117]
PEMC	*E. coli* (O157:H7)	10 cells/mL	~50 min	Campbell and Mutharasan [118]
Microcantilever array biosensor	*E. coli O157:H7*, *V. parahaemolyticus*, *Salmonella*, *S. aureus*, *L. monocytogenes*, *Shigella*	1–9 CFU/mL	<1 h	Zheng et al. [114]
Microcantilever sensor	*Yersinia*	no data	No data	Liu et al. [115]
Nanomechanical sensor	*E. coli*	~10^3^ CFU/mL	~45 min	Mertens et al. [149]
Ultrahigh frequency mechanical resonators	*S. epidermidis*	1 CFU/mL	No data	Gil-Santos et al. [116]
Impedimetric sensor (conductive polycrystalline silicon interdigitated electrodes)	*E. coli*	3 × 10^2^ CFU/mL	<1 h	de la Rica et al. [126]
DNA-aptamer based impedance biosensor	*E. coli*	9 CFU/ mL	No data	Abdelrasoul et al. [123]
Impedance biosensor	*S. typhimurium*	7 CFU/mL	40 min	Jasim et al. [119]
Impedance biosensor	*S. typhimurium*	19 CFU/mL	1.5 h	Xue et al. [120]
Impedance biosensor	*S. typhimurium*	21 CFU/mL	50 min	Wang et al. [121]
Impedance biosensor	*E. coli*, *S. aureus*	2 CFU/ mL	30 min	Zhu et al. [125]
Machine learning-based electrochemical impedance spectroscopy (EIS)	*E. coli*	2 × 10^6^ and 2 × 10^7^ CFU/mL	No data	Xu et al. [150]
Cyclic voltammetry and electrochemical impedance spectroscopy (EIS)	*S. enteritidis*	10^3^ CFU/mL	No data	Nguyen et al. [128]
Electrochemical impedance spectroscopy (EIS)	*S. epidermidis*	10^3^–10^7^ CFU/mL	No data	Golabi et al. [124]
Impedimetric paper-based biosensor	Bacterial cultures from sewage sludge	1.9 × 10^3^ CFU/mL	45 min	Rengaraj et al. [151]
Impedance biosensor with immunomagnetic separation *	*S. typhimurium*	10^2^ CFU/mL	2 h	Wang et al. [152]
MEMS-based impedance biosensor	*E. coli O157:H7*, *S. typhimurium*	10 CFU/mL	1 h	Abdullah et al. [153]
Electrochemical biosensor, cyclic voltammetry	*E. coli*	1 CFU/mL	No data	Zuser et al. [131]
Voltammetric biosensor	*L. monocytogenes* and *S. aureus*	9 CFU/mL for *L. monocytogenes* and 3 CFU/ mL for *S. aureus*	1 min	Eissa and Zaurob [132]
Voltammetric biosensor	*S. aureus*	3–5 CFU/mL	0.5 h	Farooq et al. [129]
Potentiometric immunosensor	*S. typhimurium*	5 CFU/mL	<1 h	Silva et al. [154]
Amperometric biosensor	*E. coli O157:H7*	1 CFU/mL	1 h	Dhull et al. [135]
Electrochemical biosensor	*P. aeruginosa*	1.8 × 10^−6^ mol dm^−3^	5 s	Özcan et al. [155]
Electrochemical biosensor	*E. coli*, *S. enteritidis*, *L. innocua*, *P. aeruginosa* and *S. pneumoniae*	50 CFU/mL	100 min	Feng et al. [156]
Electrochemical biosensor	*Gram-negative bacteria with Toll-like receptor 4*	368 nM	1 s	Hicks et al. [157]
Electrochemical aptasensor	*S. typhimurium*	80 CFU/mL	2 h	Wang et al. [122]
Electrochemical TLR2/6 biosensors	*B. licheniformis*, *E. hirae*	10^2^ CFU/mL (*B. licheniformis*), 10^4^ CFU/mL (*E. hirae*)	No data	McLeod et al. [158]
Electrochemical immuno-biosensor	*E. coli*	10^3^ CFU/mL	30 min	Mathelié-Guinlet et al. [127]
Sandwich-type electrochemical biosensor *	*E. coli O157:H7*	32 CFU/mL	120 min	Bu et al. [159]
All-electronic complementary metal oxide semiconductor (CMOS) biosensor	*Gram-positive* and *Gram-negative bacteria*	10^7^ CFU/mL	25 min	Nikkhoo et al. [160]
Ultrasensitive nanophotonic bimodal waveguide interferometer	*Gram-negative bacteria*	~10^5^ CFU/mL (28 aM)	30 min	Maldonado et al. [161]
Lectin-conjugated porous silicon-based biosensor	*E. coli* and *S. aureus*	10^3^ cells/ mL	No data	Yaghoubi et al. [100]
Bimodal waveguide interferometer (BiMW)	*P. aeruginosa* and *methicillin-resistant S. aureus*	49 and 29 CFU/mL (theoretical LOD)	12 min	Maldonado et al. [101]
High-throughput aptamer based photo-irradiation colorimetric biosensor *	*S. aureus*	81 CFU/mL	5.5 h	Yu et al. [51]
Fluorescent supramolecular biosensors (fSBs) *	*E. coli*	10^5^ CFU/mL	No data	Jeong et al. [56]
Portable fluorescent biosensing *	*E. coli O157:H7*, *L. monocytogenes*, *S. typhimurium*	10^2^, 10^3^, and 10^3^ CFU/mL	60 min	Xu et al. [53]
Fluorescent magnetic biosensor based on DNAzyme *	*E. coli O157:H7*	1.57 CFU/ mL	1.5 h	Zhou et al. [50]
Graphene-DNAzyme-based fluorescent biosensor *	*E. coli*	10^5^ CFU/mL	4 h	Liu et al. [55]
Aptamer-based fluorescent biosensor *	*S. sonnei*	10^3^ CFU/ mL	No data	Song et al. [54]
Optical biosensor with immunomagnetic separation *	*L. monocytogenes*	10^2^ CFU/mL	No data	Chen et al. [162]
Fluorescent biosensor *	*E. coli O157:H7*	14 CFU/mL	2 h	Xue et al. [52]
Quantum dot nanobead-based biosensor *	*S. typhimurium*	5 × 10^3^ CFU/mL	10 min	Hu et al. [163]
Nanobiosensor (AuNRs based sensor) *	*E. coli*, *P. aeruginosa*	10^9^ to 10^6^ CFU/mL (measured concentrations)	No data	Kaushal et al. [82]
Graphene field effect transistors (G-FET)	*S. aureus*, colistin *resistant* *A. baumannii*	10^4^ CFU/mL (*S. aureus*), 10^5^ CFU/mL (*A.baumannii*)	5 min	Kumar et al. [164]
Colorimetric biosensor (paper-based magnetic nanoparticle-peptide probe) *	*E. coli* O157:H7	12 CFU/mL (broth samples), 30–300 CFU/mL (spiked complex food matrices)	30 s	Suaifan et al. [60]
Colorimetric paper-based biosensor *	*S. aureus* (*L. monocytogenesis 19115*, *E. coli O157:H7*, *MRSA*, *C. albicans* *P. aeruginosa 15692*)	7 CFU/mL (pure broth) 40 CFU/mL (inoculated in food produces) 100 CFU/mL (environmental samples)	1 min	Suaifan et al. [35]
Optical immunosensor *	MRSA and non-MRSA bacteria (*E. coli*, *S. aureus* and *S. epidermis*)	10^3^ CFU/mL (visual observation) 29 CFU/mL (linear regression equation)	5 min	Raji et al. [58]
Colorimetric biosensor (using magnetic nanoparticles) *	*P. aeruginosa*	10^2^ CFU/mL	<1 min	Alhogail et al. [59]
	*L. monocytogenes*	2.17 × 10^2^ CFU/mL	<1 min	Alhogail et al. [61]

**Table 2 biosensors-12-00188-t002:** Summary of biosensors reported as capable of detecting antibiotic resistance.

Biosensor Type	Bacteria	Resistance	Limit of Detection	Assay Time	Ref.
Micromechanical oscillators	*E. coli* (XL1-Blue)	Kanamycin, tetracycline	100 cells on cantilever (50 pg/Hz)	1 h (active growth) 2 h (selective growth)	Gfeller et al. [30]
Biofunctionalized silicon micropillar arrays	*E. coli* (K-12)	Gentamicin, ciprofloxacin, ampicillin, ceftriaxone, sulfamethoxazol-trimethoprim (1:19)	10^3^ cell/ml	2–3 h	Leonard et al. [173]
Carbon screen-printed electrochemical sensor (EIS)	blaNDM (found in *Enterobacteriaceae*)	-	200 nM blaNDM	No data	Obaje et al. [31]
QCM-D (with phage spheroids)	MRSA, MSSA (*S.aureus*), *B. anthracis*, *S. typhmurium*, *S. flexneri*, *Y. enterocolitica*, *K.pneumoniae*, *B. substilis*	Penicillin-binding protein antibody latex beads	10^4^ CFU/mL (phage capture in case of *S. aureus*)	16 min/sample (spheroid-bacteria time-to-answer)	Guntupalli et al. [184]
Magnesium zinc oxide (MZO) nanostructuremodified QCM (MZO_nano_-QCM)	*E. coli* and *S. cerevisiae*	Ampicillin and tetracycline (*E. coli*), amphotericin B and miconazole (*S. cerevisiae*)	4.8 × 10^4^ to 0.9 × 10^4^ CFU/mL (for ampicillin sensitive *E. coli*) 6.1 × 10^4^ to 5.9 × 10^4^ CFU/mL (resistant *E. coli*)	10 min	Reyes et al. [172]
Electrochemical impedance spectroscopy (EIS)	*E. coli* K-12	Kanamycin, tetracyclin, erythromycin	7.1 × 10^3^ CFU/mL	~20 min	Saucedo et al. [176]
Bead-based biosensor via fluorescence imaging	*E. coli* (ATCC 25922 and 6937)	Ceftazidime, levofloxacin	5 × 10^4^ CFU/ mL	60 min	Sabhachandan et al. [179]
LSPR	*P. aeruginosa*, *E. coli*	Ceftazidime, cefotaxime, ampicillin, amoxicillin, levofloxacin, doxycycline	0.01 μg/mL in tap water, 0.5 μg/mL in 5% human serum (ceftazidimine). 10^5^ CFU/mL (used bacteria concentration)	3 h	Nag et al. [178]

## Data Availability

All relevant data are available in the manuscript.

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
