# Peer review of "Review of Label-Free Monitoring of Bacteria: From Challenging Practical Applications to Basic Research Perspectives"

_biosensors, 2022, doi:10.3390/bios12040188_

Round 1
Reviewer 1 Report
The manuscript entitled " Review of label-free monitoring of bacteria: from challenging practical applications to basic research perspectives" presents a systematic review of the present status of biosensors in bacteria monitoring. In my opinion, this study is interesting and it is within the scope of the Biosensor Journal. I have some concerns about this paper and I recommend its publication after minor revisions. Please see my comments below.
- The Abstract:
It is too lengthy and revisions are needed to make it more concise.
- The title of Section 2:
“Measure” is ambiguous, detection is more suitable
- In section 2.1.1:
Some label-based methods are also reviewed. Should be focused on label-free
- In Section 2.1:
It states that “Dielectrophoresis is often used with biosensors employing microfluidics”. However, no reference papers are reviewed.
- Section 2.2:
More papers, related with methods for antimicrobial testing should be reviewed
- Section 2.4:
The title is “STRATEGIES TO REDUCE AND CONTROL BACTERIAL ADHESION ON SURFACES”. It seems that it is irrelevant to biosensing of bacterial. Please modify it.
Author Response
1. The Abstract: It is too lengthy and revisions are needed to make it more concise.
AR: We deleted the irrelevant sentences from the abstract.
2. The title of Section 2: “Measure” is ambiguous, detection is more suitable
AR: The Reviewer is right, we changed the title to “detection”.
3. In section 2.1.1: Some label-based methods are also reviewed. Should be focused on label-free.
AR: The Reviewer is right, we shortened these parts and deleted some sentences about label-based methods.
4. In Section 2.1: It states that “Dielectrophoresis is often used with biosensors employing microfluidics”. However, no reference papers are reviewed.
AR: Thank you for this observation. We added additional references to this sentence.
5. Section 2.2: More papers, related with methods for antimicrobial testing should be reviewed
AR: We extended this section with some recent studies.
6. Section 2.4: The title is “STRATEGIES TO REDUCE AND CONTROL BACTERIAL ADHESION ON SURFACES”. It seems that it is irrelevant to biosensing of bacterial. Please modify it.
AR: Thank you for noticing this. We changed the title to STRATEGIES TO REDUCE AND CONTROL BACTERIAL ADHESION ON SENSING SURFACES. With this title we highlight the importance to create sensing surfaces with bacteria repellent and adhesive properties. We added a recent article to the text which is relevant and up-to-date in this topic.
Reviewer 2 Report
The Authors review the literature on the label-free biosensors for bacteria. The in-depth literature review is exhaustive and well-written. It helps the reader to get the guidelines for the study and use of biosensors for monitoring of bacteria. I suggest the manuscript publication in the Biosensors Journal, after addressing the following comments.
- In the Section 1, the description of the bacteria morphology is well exhaustive. However, the Author should report the reference and a sketch of the bacterium cross-section. Moreover, the Authors should report the optical and electrical properties of the bacteria components (see as example Stiffness of optical traps: quantitative agreement between experiment and electromagnetic theory. Physical review letters, 95(16), 168102, 2005; Dielectric properties of E. coli cell as simulated by the three-shell spheroidal model. Biophysical chemistry, 122(2), 136-142, 2006; Non-invasive determination of bacterial single cell properties by electrorotation. Biochimica et Biophysica Acta (BBA)-Molecular Cell Research, 1450(1), 53-60, 1999).
- In the Section 2, the term “Measuring” in the title is misleading. In the end of the section, the Authors should provide a brief summary on the advantages/disadvantages of the main techniques, i.e. optical, mechanical, electrochemical,... A table could help the reader to rate the performance of each technology.
- The phenomenon of the Anti Microbial Resistance represents a hazard for all the world. To counteract it, several devices have been proposed in literature, in order to monitor a single bacterium (Monitoring of individual bacteria using electro-photonic traps. Biomedical Optics Express, 10(7), 3463-3471. 2019; Single-cell bacterium identification with a SOI optical microcavity. Applied Physics Letters, 109(13), 133510. 2016; Cavity-enhanced optical trapping of bacteria using a silicon photonic crystal. Lab on a Chip, 13(22), 4358-4365, 2013) or a biofilm (Attachment and antibiotic response of early-stage biofilms studied using resonant hyperspectral imaging. NPJ biofilms and microbiomes, 6(1), 1-7., 2020; Novel micro-nano optoelectronic biosensor for label-free real-time biofilm monitoring. Biosensors, 11(10), 361, 2021; An integrated microsystem for real-time detection and threshold-activated treatment of bacterial biofilms. ACS applied materials & interfaces, 9(37), 31362-31371, 2017). The Authors should go into detail of this topic.
Author Response
1. In the Section 1, the description of the bacteria morphology is well exhaustive. However, the Author should report the reference and a sketch of the bacterium cross-section. Moreover, the Authors should report the optical and electrical properties of the bacteria components (see as example Stiffness of optical traps: quantitative agreement between experiment and electromagnetic theory. Physical review letters, 95(16), 168102, 2005; Dielectric properties of E. coli cell as simulated by the three-shell spheroidal model. Biophysical chemistry, 122(2), 136-142, 2006; Non-invasive determination of bacterial single cell properties by electrorotation. Biochimica et Biophysica Acta (BBA)-Molecular Cell Research, 1450(1), 53-60, 1999).
AR: Thank you for this observation. We report the optical and electrical properties of the bacteria components, and added the suggested references to text.
2. In the Section 2, the term “Measuring” in the title is misleading. In the end of the section, the Authors should provide a brief summary on the advantages/disadvantages of the main techniques, i.e. optical, mechanical, electrochemical,... A table could help the reader to rate the performance of each technology.
AR: The Reviewer is right, we changed the title to “detection”. In the end of the section, a brief text is inserted about the advantages/disadvantages of the main techniques.
3. The phenomenon of the Anti Microbial Resistance represents a hazard for all the world. To counteract it, several devices have been proposed in literature, in order to monitor a single bacterium (Monitoring of individual bacteria using electro-photonic traps. Biomedical Optics Express, 10(7), 3463-3471. 2019; Single-cell bacterium identification with a SOI optical microcavity. Applied Physics Letters, 109(13), 133510. 2016; Cavity-enhanced optical trapping of bacteria using a silicon photonic crystal. Lab on a Chip, 13(22), 4358-4365, 2013) or a biofilm (Attachment and antibiotic response of early-stage biofilms studied using resonant hyperspectral imaging. NPJ biofilms and microbiomes, 6(1), 1-7., 2020; Novel micro-nano optoelectronic biosensor for label-free real-time biofilm monitoring. Biosensors, 11(10), 361, 2021; An integrated microsystem for real-time detection and threshold-activated treatment of bacterial biofilms. ACS applied materials & interfaces, 9(37), 31362-31371, 2017). The Authors should go into detail of this topic.
AR: We wrote additional sentences to this part and added the mentioned references to the text.
Round 2
Reviewer 2 Report
The Authors have satisfactory replied to the Reviewer comments, by improving the manuscript. The manuscript publication is endorsed.